# Modeling GPR Wave Propagation in Complex Underground Structures Using Conformal ADI-FDTD Algorithm

**Yinping Li [1], Niannian Wang [1], Jianwei Lei [1,*], Fuming Wang [1] and Ce Li [2]**

1   School of Water Conservancy Engineering, Zhengzhou University, Zhengzhou 450001, China; lyp07170111@163.com (Y.L.); wnnian@zzu.edu.cn (N.W.); fuming573@126.com (F.W.)
2   School of Mechanical Electronic & Information Engineering, China University of Mining & Technology, Beijing 100083, China; celi@cumtb.edu.cn
*   Correspondence: leijianwei@gs.zzu.edu.cn; Tel.: +86-153-3382-7823

**Abstract:** Ground Penetrating Radar (GPR) is a shallow geophysical method for detecting and locating subsurface targets. The GPR image echo characteristics of complex underground structures can be obtained by carrying out GPR forward modeling research. The traditional finite-difference time-domain (FDTD) method has low efficiency and accuracy. The alternating direction implicit FDTD (ADI-FDTD) algorithm surmounts the stability limitations of the traditional FDTD method, making it possible to select a larger time step for higher computational efficiency. For circular underground structures, a pseudowave produced by the ladder approximation method can be corrected using the surface conformal technique. This paper proposes a high-efficiency and high-accuracy GPR forward modeling method that combines the ADI-FDTD algorithm and surface conformal technology. The performance of the conformal ADI-FDTD algorithm is verified by a simple two-layer model. Based on the proposed algorithm, the GPR image features of three complex underground structure models are obtained. Finally, a field experiment is used to support the accuracy and usefulness of the conformal ADI-FDTD algorithm. The numerical simulation results and experimental results show that the conformal ADI-FDTD algorithm reduces the pseudodiffraction wave caused by the ladder approximation method and can significantly improve the computing efficiency for complex underground structure models.

**Keywords:** GPR; ADI-FDTD; surface conformal technology; complex underground structure; numerical simulations

## 1. Introduction

With rapid urban development, the scale of underground pipeline networks has gradually expanded [1,2]. Pipelines with different functions and purposes are arranged alternately, and the arrangement of the underground pipeline network is very complicated. As detailed distribution information for the entire underground pipeline network cannot be accurately obtained, accidents (e.g., the collapse of old pipelines or the cutting-off of existing pipelines) often occur [3–6]. In addition, road collapse accidents due to pipeline leaks occur frequently. The quick and accurate determination of the type, size, and location of underground pipelines and hidden diseases around underground pipelines has become necessary in urban road construction and maintenance projects.

GPR is widely regarded as one of the most powerful and useful geophysical methods [7]. GPR has the advantage of non-destructive detection and is fast, lightweight, and easy to operate with strong anti-interference and high resolution, etc. compared with other Non-Destructive Testing (NDT) methods such as seismic, sonic, temperature, resistivity, logging, and magnetic exploration methods [8,9]. Information about the location, size, and dielectric properties of complex underground structures can be obtained by inversion analysis of the GPR echo signals [10–13]. Through accurate and efficient GPR forward modeling of complex underground structure models, the propagation law of GPR

electromagnetic waves in underground structures can be obtained [14–16]. This lays the theoretical foundation for inversion analysis and helps to improve the interpretation and processing accuracy of GPR-measured data. At present, the most commonly used GPR forward simulation techniques include the finite element method (FEM) [17,18], the ray tracing method (RTM) [19,20], the method of moment (MOM) [21], the finite difference time domain method (FDTD) [22–24], the pseudospectral time domain method (PSTD) [25], and the symplectic algorithm [26–28], among others. Although research into GPR forward simulation has achieved fruitful results, these algorithms still have some limitations in terms of computational accuracy and efficiency. For example, the FEM may appear as a "pseudosolving" phenomenon during calculation; the RTM cannot consider the dynamic characteristics of the electromagnetic wave of the ground penetrating radar, while the time step of the symplectic algorithm needs to satisfy the Courant–Friedrichs–Lewy (CFL) stability condition, and the computational efficiency is limited. Therefore, it is necessary to propose a GPR forward modeling method with high computational accuracy and fast computational efficiency.

The ADI-FDTD algorithm overcomes the CFL stability condition and can employ a larger time step to improve the computing simulation efficiency [29–31]. The efficiency and accuracy of GPR forward simulation are also related to the modeling method. Some methods are available to improve the accuracy of GPR simulation calculations for circular structures, such as sub-grid technology and conformal grid technology. The sub-grid technique has been widely used to improve the calculation accuracy, but it generates pseudowaves at the interface of the fine and coarse grids [32,33]. The derivations and calculations for conformal grid technology are simple, making it very suitable for circular underground pipes [34–36].

In this paper, the efficient and accurate GPR forward models are established to simulate GPR electromagnetic wave propagation in underground structures, employing the ADI-FDTD algorithm and surface conformal technology. We obtained the reflection profile images of multi-pipe models and complex underground pipe models with hidden diseases. By analyzing the profile images, the spatial propagation characteristics of radar waves can be more clearly understood, and the interpretation accuracy of the data can be improved. A field experiment proved the correctness and effectiveness of the proposed algorithm in actual detection. The numerical simulation results and field experiment indicate that the conformal ADI-FDTD method requires less computation time and can greatly reduce the pseudowave effect compared with the traditional FDTD.

## 2. Methodology

### 2.1. ADI-FDTD Algorithm

For transverse magnetic (TM) waves in two-dimensional lossy media, the Maxwell equations [37] are expressed as

$$\mu \frac{\partial H_x}{\partial t} = -\frac{\partial E_z}{\partial y}, \tag{1}$$

$$\mu \frac{\partial H_y}{\partial t} = \frac{\partial E_z}{\partial x}, \tag{2}$$

$$\varepsilon \frac{\partial E_z}{\partial t} + \sigma E_z = \frac{\partial H_y}{\partial x} - \frac{\partial H_x}{\partial y}, \tag{3}$$

where $H_x$ and $H_y$ are the magnetic field intensities in the $x$ and $y$ directions; $E_z$ is the electric field intensity in the $z$ direction; $\sigma$ is the electrical conductivity; $\varepsilon$ is the dielectric constant; $\mu$ is the magnetic permeability.

In the ADI-FDTD algorithm, we divide a time step into two sub-steps and discretize them in different ways. In the sub-step $n \rightarrow n + 1/2$, the $x$-direction derivative adopts the implicit difference format, and the $y$-direction derivative adopts the explicit difference format. In the sub-step of $n + 1/2 \rightarrow n + 1$, the $x$-direction derivative adopts the explicit difference format, and the $y$-direction derivative adopts the implicit difference format.

In the $n \to n + 1/2$ time step, the difference scheme of Equation (1) is as follows:

$$H_x^{n+\frac{1}{2}}(i, j + \tfrac{1}{2}) = H_x^n(i, j + \tfrac{1}{2}) - \frac{\Delta t}{2\mu} \cdot \frac{1}{\Delta y} \cdot [E_z^n(i, j + 1) - E_z^n(i, j)]. \tag{4}$$

Similarly, the difference discretization scheme of Equation (2) is

$$H_y^{n+\frac{1}{2}}(i + \tfrac{1}{2}, j) = H_y^n(i + \tfrac{1}{2}, j) + \frac{\Delta t}{2\mu} \cdot \frac{1}{\Delta x} \cdot \left[ E_z^{n+\frac{1}{2}}(i + 1, j) - E_z^{n+\frac{1}{2}}(i, j) \right]. \tag{5}$$

For Equation (3), the $E_z$ component takes the mean value of time $n\Delta t$ and time $(n + 1/2)\Delta t$, where the difference discretization scheme of Equation (3) is

$$E_z^{n+\frac{1}{2}}(i, j) = CA \cdot E_z^n(i, j) + CB \cdot \left\{ \frac{1}{\Delta x} \cdot \left[ H_y^{n+\frac{1}{2}}(i + \tfrac{1}{2}, j) - H_y^{n+\frac{1}{2}}(i - \tfrac{1}{2}, j) \right] \right.$$
$$\left. - \frac{1}{\Delta y} \cdot \left[ H_x^n(i, j + \tfrac{1}{2}) - H_x^n(i, j - \tfrac{1}{2}) \right] \right\} \tag{6}$$

where

$$CA = \frac{4\varepsilon - \sigma\Delta t}{4\varepsilon + \sigma\Delta t} \quad CB = \frac{2\Delta t}{4\varepsilon + \sigma\Delta t}. \tag{7}$$

Equations (4)–(6) are the time domain advancing formulas of the electromagnetic field from step $n \to n + 1/2$. Equation (4) is called the display format because the right side of the formula only involves the field value at time $n$. However, both sides of Equations (5) and (6) include the simultaneous field value, which is called the implicit format, which cannot be used to calculate the display time advance. To solve this problem, we substitute Equation (5) into Equation (6) and eliminate $H_y^{n+1/2}$, and we can derive

$$-\frac{\Delta t}{2\mu\Delta x^2} CB \cdot E_z^{n+\frac{1}{2}}(i - 1, j) + \left[ 1 + \frac{\Delta t}{\mu\Delta x^2} \cdot CB \right] \cdot E_z^{n+\frac{1}{2}}(i, j) - \frac{\Delta t}{2\mu\Delta x^2} \cdot CB \cdot E_z^{n+\frac{1}{2}}(i + 1, j)$$
$$= CA \cdot E_z^n(i, j) + \frac{CB}{\Delta x} \cdot \left[ H_y^n(i + \tfrac{1}{2}, j) - H_y^n(i - \tfrac{1}{2}, j) \right] - \frac{CB}{\Delta y} \cdot \left[ H_x^n(i, j + \tfrac{1}{2}) - H_x^n(i, j - \tfrac{1}{2}) \right], \tag{8}$$

where

$$a_i = -\frac{\Delta t}{2\mu\Delta x^2} \cdot CB \quad b_i = 1 + \frac{\Delta t}{\mu\Delta x^2} \cdot CB \quad c_i = -\frac{\Delta t}{2\mu\Delta x^2} \cdot CB$$
$$d_i = CA \cdot E_z^n(i, j) + \frac{CB}{\Delta x} \cdot \left[ H_y^n(i + \tfrac{1}{2}, j) - H_y^n(i - \tfrac{1}{2}, j) \right] - \frac{CB}{\Delta y} \cdot \left[ H_x^n(i, j + \tfrac{1}{2}) - H_x^n(i, j - \tfrac{1}{2}) \right]. \tag{9}$$

Then, Equation (8) can be expressed as

$$a_i E_z^{n+\frac{1}{2}}(i - 1, j) + b_i E_z^{n+\frac{1}{2}}(i, j) + c_i E_z^{n+\frac{1}{2}}(i + 1, j) = d_i, \tag{10}$$

and Equation (9) is written in matrix form as

$$AX = Y, \tag{11}$$

where

$$X = \begin{bmatrix} E_z^{n+\frac{1}{2}}(1, j) \\ \vdots \\ E_z^{n+\frac{1}{2}}(i, j) \\ \vdots \\ E_z^{n+\frac{1}{2}}(i_{max}, j) \end{bmatrix}, Y = \begin{bmatrix} d_1 \\ \vdots \\ d_i \\ \vdots \\ d_{imax} \end{bmatrix}, A = \begin{bmatrix} b_1 & c_1 & 0 & 0 & 0 \\ a_2 & b_2 & c_2 & 0 & 0 \\ 0 & \ddots & \ddots & \ddots & 0 \\ 0 & 0 & a_{imax-1} & b_{imax-1} & c_{imax-1} \\ 0 & 0 & 0 & a_{imax} & b_{imax} \end{bmatrix}. \tag{12}$$

Here, the matrix $A$ is a tridiagonal strip matrix, and the first-row elements ($b_1$ and $c_1$) and the last-row elements ($a_{imax}$ and $b_{imax}$) are given by absorption boundary conditions.

In the $n + 1/2 \rightarrow n + 1$ time step, the difference discretization schemes of Equations (1)–(3) are

$$H_x^{n+1}\left(i, j + \frac{1}{2}\right) = H_x^{n+\frac{1}{2}}\left(i, j + \frac{1}{2}\right) - \frac{\Delta t}{2\mu} \cdot \frac{1}{\Delta y} \cdot \left[E_z^{n+1}(i, j + 1) - E_z^{n+1}(i, j)\right], \tag{13}$$

$$H_y^{n+1}\left(i + \frac{1}{2}, j\right) = H_y^{n+\frac{1}{2}}\left(i + \frac{1}{2}, j\right) + \frac{\Delta t}{2\mu} \cdot \frac{1}{\Delta x} \cdot \left[E_z^{n+\frac{1}{2}}(i + 1, j) - E_z^{n+\frac{1}{2}}(i, j)\right], \tag{14}$$

$$E_z^{n+1}(i, j) = CA \cdot E_z^{n+\frac{1}{2}}(i, j) + CB \cdot \left\{\frac{1}{\Delta x} \cdot \left[H_y^{n+\frac{1}{2}}\left(i + \frac{1}{2}, j\right) - H_y^{n+\frac{1}{2}}\left(i - \frac{1}{2}, j\right)\right]\right.$$
$$\left. - \frac{1}{\Delta y} \cdot \left[H_x^{n+1}\left(i, j + \frac{1}{2}\right) - H_x^{n+1}\left(i, j - \frac{1}{2}\right)\right]\right\}. \tag{15}$$

The time domain advance of the electromagnetic field in sub-step $n + 1/2 \rightarrow n + 1$ can be calculated by Equations (13)–(15), as Equations (13) and (15) contain the simultaneous field value (i.e., implicit format) and cannot be used to calculate the display time advance. From Equations (13) and (15), we can derive

$$-\frac{\Delta t}{2\mu\Delta y^2} \cdot CB \cdot E_z^{n+1}(i, j - 1) + \left[1 + \frac{\Delta t}{\mu\Delta y^2} \cdot CB\right] \cdot E_z^{n+1}(i, j) - \frac{\Delta t}{2\mu\Delta y^2} \cdot CB \cdot E_z^{n+1}(i, j + 1) =$$
$$CA \cdot E_z^{n+\frac{1}{2}}(i, j) - \frac{CB}{\Delta y} \cdot \left[H_x^{n+\frac{1}{2}}\left(i, j + \frac{1}{2}\right) - H_x^{n+\frac{1}{2}}\left(i, j - \frac{1}{2}\right)\right] + \frac{CB}{\Delta x} \cdot \left[H_y^{n+\frac{1}{2}}\left(i + \frac{1}{2}, j\right) - H_y^{n+\frac{1}{2}}\left(i - \frac{1}{2}, j\right)\right], \tag{16}$$

where

$$a_j = -\frac{\Delta t}{2\mu\Delta y^2} \cdot CB \quad b_j = 1 + \frac{\Delta t}{\mu\Delta y^2} \cdot CB \quad c_j = -\frac{\Delta t}{2\mu\Delta y^2} \cdot CB$$
$$d_j = CA \cdot E_z^{n+\frac{1}{2}}(i, j) - \frac{CB}{\Delta y} \cdot \left[H_x^{n+\frac{1}{2}}\left(i, j + \frac{1}{2}\right) - H_x^{n+\frac{1}{2}}\left(i, j - \frac{1}{2}\right)\right] + \frac{CB}{\Delta x} \cdot \left[H_y^{n+\frac{1}{2}}\left(i + \frac{1}{2}, j\right) - H_y^{n+\frac{1}{2}}\left(i - \frac{1}{2}, j\right)\right]. \tag{17}$$

Then, Equation (16) can be rewritten as

$$a_j E_z^{n+1}(i, j - 1) + b_j E_z^{n+1}(i, j) + c_j E_z^{n+1}(i, j + 1) = d_j, \tag{18}$$

and Equation (17) can be written in the following matrix form:

$$AX = Y, \tag{19}$$

where

$$X = \begin{bmatrix} E_z^{n+1}(i, 1) \\ \vdots \\ E_z^{n+1}(i, j) \\ \vdots \\ E_z^{n+1}(i, j_{max}) \end{bmatrix}, Y = \begin{bmatrix} d_1 \\ \vdots \\ d_j \\ \vdots \\ d_{jmax} \end{bmatrix}, A = \begin{bmatrix} b_1 & c_1 & 0 & 0 & 0 \\ a_2 & b_2 & c_2 & 0 & 0 \\ 0 & \ddots & \ddots & \ddots & 0 \\ 0 & 0 & a_{jmax-1} & b_{jmax-1} & c_{jmax-1} \\ 0 & 0 & 0 & a_{jmax} & b_{jmax} \end{bmatrix}. \tag{20}$$

Here, matrix $A$ is a tridiagonal strip matrix, and the first-row elements ($b_1$ and $c_1$) and the last-row elements ($a_{jmax}$ and $b_{jmax}$) are given by absorption boundary conditions.

In conclusion, the steps for calculating the 2D TM wave by ADI-FDTD are as follows:

In the $n \rightarrow n + 1/2$ time step:

(1) Calculate $H_x^{n+1/2}$ by Equation (4);
(2) Calculate $E_z^{n+1/2}$ by Equation (10);
(3) Calculate $H_y^{n+1/2}$ by Equation (5).

In the $n + 1/2 \rightarrow n + 1$ time step:

(1) Calculate $H_y^{n+1}$ by Equation (14);
(2) Calculate $E_z^{n+1}$ by Equation (18);
(3) Calculate $H_x^{n+1}$ by Equation (13).

### 2.2. Surface Conformal Technology

The surface conformal technique was used to simulate the mesh generation of circular underground structures. Figure 1 shows schematic diagrams of the grid division of the circular structure model constructed by different methods. Figure 1a shows the actual subdivision grid of the circular structure, where the white squares are ordinary and the red squares are the actual subdivisions. Figure 1b shows the conventional ladder approximation subdivision grid for a circular structure, where the green squares are the subdivisions obtained by the step approximation method. The subdivision grid of a circular structure based on surface conformal technology is shown in Figure 1c, where the orange squares are conformal areas and the blue squares are non-conformal areas.

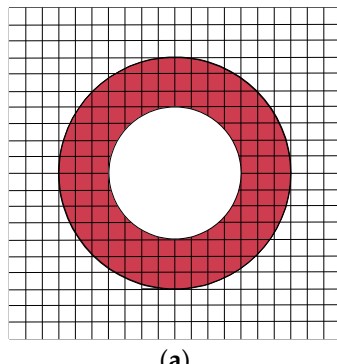
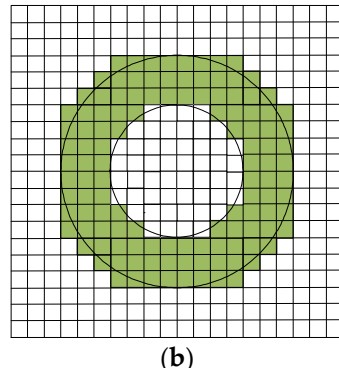
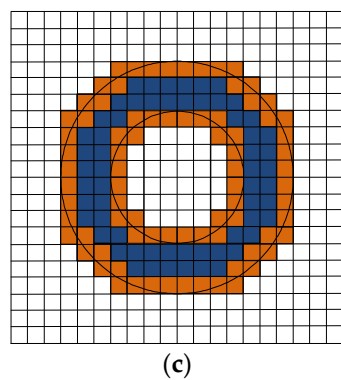

(a)      (b)      (c)

**Figure 1.** Grid dissection of the circular structure: (**a**) Actual grid dissection of a circular structure; (**b**) Grid dissection of circular structure by ladder approximation method; (**c**) Dissection of circular structure by conformal grid technology.

A conformal grid point in Figure 1c is taken as an example in order to demonstrate the selection of equivalent medium parameters for the conformal grid region in 2D TM waves. As shown in Figure 2, where $F$ is the sampling point of the electric field $E_z$; $A$, $B$ and $C$, $D$ are sampling points of the magnetic field $H_x$ and $H_y$, respectively; $\Delta y$ and $\Delta x$ are the height and width of the grid, respectively; $S_{xy1}$ and $S_{xy2}$ are the areas of media 1 and 2 in the conformal grid, respectively; $L_{x1}$ and $L_{x2}$ are the lengths occupied by media 1 and 2 on the edge of magnetic field node $B$, respectively; and $L_{y1}$ and $L_{y2}$ are the lengths occupied by media 1 and 2 on the edge of magnetic field node $D$, respectively.

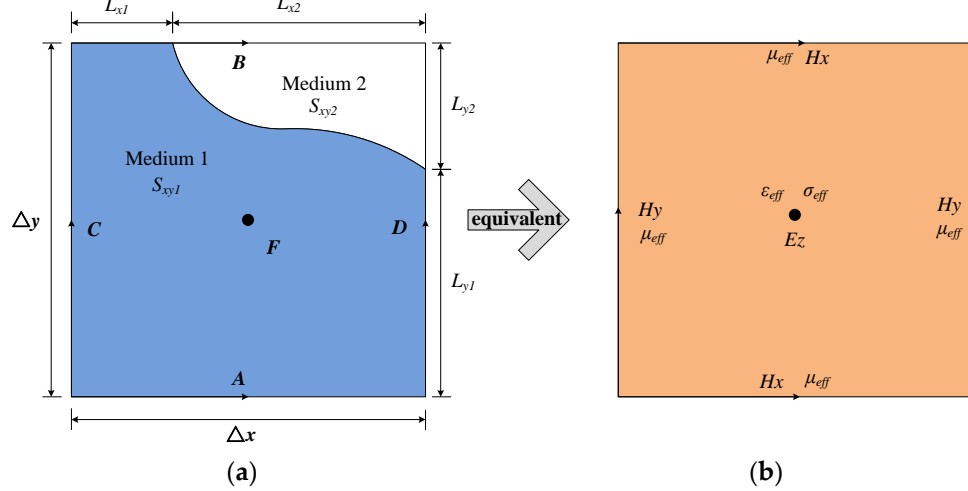

(a)      (b)

**Figure 2.** Equivalent schematic of the media parameters at the conformal grid points: (**a**) Actual grids; (**b**) Conformal grids.

Considering the two-dimensional TM wave, for the conformal grid of an ideal conductor (metal conductor), the recursive formula of a magnetic field node remains unchanged and is still calculated according to conventional ADI-FDTD. For the recursive formula of the electric field node, we make the following changes:

In the $n \to n + 1/2$ time step, Equation (6) is changed to

$$
\begin{aligned}
E_z^{n+\frac{1}{2}}(i,j) = CA \cdot E_z^n(i,j) + CB \cdot \bigg\{ & \frac{l_y}{S} \cdot \left[ H_y^{n+\frac{1}{2}}\left(i+\tfrac{1}{2},j\right) - H_y^{n+\frac{1}{2}}\left(i-\tfrac{1}{2},j\right) \right] \\
& - \frac{l_x}{S} \cdot \left[ H_x^n\left(i,j+\tfrac{1}{2}\right) - H_x^n\left(i,j-\tfrac{1}{2}\right) \right] \bigg\}.
\end{aligned}
\tag{21}
$$

where $l_x$ and $l_y$ represent the length of the corresponding edge of the electric field node outside the conductor, and $S$ represents the area of the cell outside the conductor.

We substitute $H_y^{n+1/2}$ in Equation (5) into Equation (21) and obtain the update equation about $E_z^{n+1/2}$

$$
a_i' E_z^{n+\frac{1}{2}}(i-1,j) + b_i' E_z^{n+\frac{1}{2}}(i,j) + c_i' E_z^{n+\frac{1}{2}}(i+1,j) = d_i',
\tag{22}
$$

where

$$
\begin{aligned}
a_i' = -\frac{\Delta t}{2\mu\Delta x} \cdot \frac{l_y}{S} \cdot CB \quad & b_i' = 1 + \frac{\Delta t}{\mu\Delta x} \cdot \frac{l_y}{S} \cdot CB \quad c_i' = -\frac{\Delta t}{2\mu\Delta x} \cdot \frac{l_y}{S} \cdot CB \\
d_i' = CA \cdot E_z^n(i,j) + CB \cdot \frac{l_y}{S} \cdot & \left[ H_y^n\left(i+\tfrac{1}{2},j\right) - H_y^n\left(i-\tfrac{1}{2},j\right) \right] - CB \cdot \frac{l_x}{S} \cdot \left[ H_x^n\left(i,j+\tfrac{1}{2}\right) - H_x^n\left(i,j-\tfrac{1}{2}\right) \right].
\end{aligned}
\tag{23}
$$

In the $n + 1/2 \to n + 1$ time step, the difference discretization schemes of Equation (15) are

$$
\begin{aligned}
E_z^{n+1}(i,j) = CA \cdot E_z^{n+\frac{1}{2}}(i,j) + CB \cdot \bigg\{ & \frac{l_y}{S} \cdot \left[ H_y^{n+\frac{1}{2}}\left(i+\tfrac{1}{2},j\right) - H_y^{n+\frac{1}{2}}\left(i-\tfrac{1}{2},j\right) \right] \\
& - \frac{l_x}{S} \cdot \left[ H_x^{n+1}\left(i,j+\tfrac{1}{2}\right) - H_x^{n+1}\left(i,j-\tfrac{1}{2}\right) \right] \bigg\},
\end{aligned}
\tag{24}
$$

Similarly, we can obtain the iterative format of $E_z^{n+1}$

$$
a_j' E_z^{n+1}(i,j-1) + b_j' E_z^{n+1}(i,j) + c_j' E_z^{n+1}(i,j+1) = d_j',
\tag{25}
$$

where

$$
\begin{aligned}
a_j' = -\frac{\Delta t}{2\mu\Delta y} \cdot \frac{l_x}{S} \cdot CB \quad & b_j' = 1 + \frac{\Delta t}{\mu\Delta y} \cdot \frac{l_x}{S} \cdot CB \quad c_j' = -\frac{\Delta t}{2\mu\Delta y} \cdot \frac{l_x}{S} \cdot CB \\
d_j' = CA \cdot E_z^{n+\frac{1}{2}}(i,j) - CB \cdot \frac{l_x}{S} \cdot & \left[ H_x^{n+\frac{1}{2}}\left(i,j+\tfrac{1}{2}\right) - H_x^{n+\frac{1}{2}}\left(i,j-\tfrac{1}{2}\right) \right] + CB \cdot \frac{l_y}{S} \cdot \left[ H_y^{n+\frac{1}{2}}\left(i+\tfrac{1}{2},j\right) - H_y^{n+\frac{1}{2}}\left(i-\tfrac{1}{2},j\right) \right].
\end{aligned}
\tag{26}
$$

For media (non-metallic conductor) surface conformal meshes, the permittivity, permeability, and conductivity of dielectrics 1 and 2 are assumed to be $\varepsilon_1$, $\mu_1$, and $\sigma_1$ and $\varepsilon_2$, $\mu_2$, and $\sigma_2$, respectively. A weighted average of the electromagnetic coefficients, according to the sizes of the conformal grids occupied by media 1 and 2, respectively, provides the following equivalent medium parameters for the dielectric constant, permeability, and conductivity:

$$
\varepsilon_{eff} = \frac{L_{x1}\varepsilon_1 + L_{x2}\varepsilon_2}{\delta}, \; \sigma_{eff} = \frac{L_{x1}\sigma_1 + L_{x2}\sigma_2}{\delta}, \; \mu_{eff} = \frac{S_{xy1}\mu_1 + S_{xy2}\mu_2}{\delta^2}.
\tag{27}
$$

In the $n \to n + 1/2$ time step, the difference discretization schemes of the electric field node and the magnetic field node are written as

$$
H_x^{n+\frac{1}{2}}\left(i,j+\frac{1}{2}\right) = H_x^n\left(i,j+\frac{1}{2}\right) - \frac{\Delta t}{2\mu_{eff}} \cdot \frac{1}{\Delta y} \cdot \left[ E_z^n(i,j+1) - E_z^n(i,j) \right],
\tag{28}
$$

$$H_y^{n+\frac{1}{2}}\left(i+\frac{1}{2},j\right) = H_y^n\left(i+\frac{1}{2},j\right) + \frac{\Delta t}{2\mu_{eff}} \cdot \frac{1}{\Delta x} \cdot \left[E_z^{n+\frac{1}{2}}(i+1,j) - E_z^{n+\frac{1}{2}}(i,j)\right], \quad (29)$$

$$a_i E_z^{n+\frac{1}{2}}(i-1,j) + b_i E_z^{n+\frac{1}{2}}(i,j) + c_i E_z^{n+\frac{1}{2}}(i+1,j) = d_i, \quad (30)$$

where

$$\begin{aligned} CA &= \frac{4\varepsilon_{eff}-\sigma_{eff}\Delta t}{4\varepsilon_{eff}+\sigma_{eff}\Delta t} \quad CB = \frac{2\Delta t}{4\varepsilon_{eff}+\sigma_{eff}\Delta t} \\ a_i &= -\frac{\Delta t}{2\mu_{eff}\Delta x^2} \cdot CB \quad b_i = 1 + \frac{\Delta t}{\mu_{eff}\Delta x^2} \cdot CB \quad c_i = -\frac{\Delta t}{2\mu_{eff}\Delta x^2} \cdot CB \\ d_i &= CA \cdot E_z^n(i,j) + \frac{CB}{\Delta x} \cdot \left[H_y^n\left(i+\frac{1}{2},j\right) - H_y^n\left(i-\frac{1}{2},j\right)\right] - \frac{CB}{\Delta y} \cdot \left[H_x^n\left(i,j+\frac{1}{2}\right) - H_x^n\left(i,j-\frac{1}{2}\right)\right]. \end{aligned} \quad (31)$$

In the $n + 1/2 \rightarrow n + 1$ time step, the difference discretization schemes of the electric field node and the magnetic field node are

$$H_x^{n+1}\left(i,j+\frac{1}{2}\right) = H_x^{n+\frac{1}{2}}\left(i,j+\frac{1}{2}\right) - \frac{\Delta t}{2\mu_{eff}} \cdot \frac{1}{\Delta y} \cdot \left[E_z^{n+1}(i,j+1) - E_z^{n+1}(i,j)\right], \quad (32)$$

$$H_y^{n+1}\left(i+\frac{1}{2},j\right) = H_y^{n+\frac{1}{2}}\left(i+\frac{1}{2},j\right) + \frac{\Delta t}{2\mu_{eff}} \cdot \frac{1}{\Delta x} \cdot \left[E_z^{n+\frac{1}{2}}(i+1,j) - E_z^{n+\frac{1}{2}}(i,j)\right], \quad (33)$$

$$a_j E_z^{n+1}(i,j-1) + b_j E_z^{n+1}(i,j) + c_j E_z^{n+1}(i,j+1) = d_j, \quad (34)$$

where

$$\begin{aligned} CA &= \frac{4\varepsilon_{eff}-\sigma_{eff}\Delta t}{4\varepsilon_{eff}+\sigma_{eff}\Delta t} \quad CB = \frac{2\Delta t}{4\varepsilon_{eff}+\sigma_{eff}\Delta t} \\ a_j &= -\frac{\Delta t}{2\mu_{eff}\Delta y^2} \cdot CB \quad b_j = 1 + \frac{\Delta t}{\mu_{eff}\Delta y^2} \cdot CB \quad c_j = -\frac{\Delta t}{2\mu_{eff}\Delta y^2} \cdot CB \\ d_j &= CA \cdot E_z^{n+\frac{1}{2}}(i,j) - \frac{CB}{\Delta y} \cdot \left[H_x^{n+\frac{1}{2}}\left(i,j+\frac{1}{2}\right) - H_x^{n+\frac{1}{2}}\left(i,j-\frac{1}{2}\right)\right] + \frac{CB}{\Delta x} \cdot \left[H_y^{n+\frac{1}{2}}\left(i+\frac{1}{2},j\right) - H_y^{n+\frac{1}{2}}\left(i-\frac{1}{2},j\right)\right]. \end{aligned} \quad (35)$$

### 2.3. UPML Absorbing Boundary Condition

GPR electromagnetic wave propagation in underground structures is an open domain problem. Thus, it is necessary to set reasonable absorbing boundary conditions at the truncated boundary of the computational region. In this paper, the uniaxial anisotropic absorption layer (UPML) with easy programming and simple iterative formula is used as the absorption boundary [38]. The Maxwell curl equation in the UPML medium can be written as:

$$\begin{aligned} \nabla \times H &= j\omega\varepsilon\overline{S} \cdot E \\ \nabla \times E &= -j\omega\mu\overline{S} \cdot H \end{aligned} \quad (36)$$

where $\overline{S}$ has the characteristics of a uniaxial anisotropic medium, expressed as:

$$\overline{S} = \begin{bmatrix} s_y s_z / s_x & 0 & 0 \\ 0 & s_z s_x / s_y & 0 \\ 0 & 0 & s_x s_y / s_z \end{bmatrix}. \quad (37)$$

Here, $s_i = \kappa_i + \sigma_i / j\omega\varepsilon_0$ ($i = x, y, z$) is the diagonal tensor of a uniaxially anisotropic medium (for a 2D GPR wave, $s_z$ is 1), $\sigma_i$ is the attenuation factor of the UPML region, and $\kappa_i$ is used to absorb the modulated wave that reaches the UPML layer.

For the 2D TM wave, $E_x = 0$, $E_y = 0$, and $H_z = 0$, and the ADI-FDTD formula under the UPML boundary conditions is shown below.

When $D_z = \varepsilon s_y E_z$, the time advance formula of $H_x$, $H_y \rightarrow D_z$ is as follows:

$$\frac{\partial H_y}{\partial x} - \frac{\partial H_x}{\partial y} = \kappa_x \frac{\partial D_z}{\partial t} + \frac{\sigma_x}{\varepsilon_0} D_z, \quad (38)$$

and the time advance formula of $D_z \rightarrow E_z$ is:

$$\frac{\partial D_z}{\partial t} = \varepsilon_1 \kappa_y \frac{\partial E_z}{\partial t} + \varepsilon_1 \frac{\sigma_y}{\varepsilon_0} E_z, \tag{39}$$

When $B_x = \mu_1 H_x / s_x$ and $B_y = \mu_1 H_y / s_y$, the time advance formulae for $E_z \rightarrow B_x$, $B_y$ are as follows:

$$\frac{\partial E_z}{\partial y} = -\kappa_y \frac{\partial B_x}{\partial t} - \frac{\sigma_y}{\varepsilon_0} B_x, \tag{40}$$

$$\frac{\partial E_z}{\partial x} = -\kappa_x \frac{\partial B_y}{\partial t} + \frac{\sigma_x}{\varepsilon_0} B_y, \tag{41}$$

and the time advance formulae for $B_x$, $B_y \rightarrow H_x$ and $H_y$ are as follows:

$$\kappa_x \frac{\partial B_x}{\partial t} + \frac{\sigma_x}{\varepsilon_0} B_x = \mu_1 \frac{\partial H_x}{\partial t}, \tag{42}$$

$$\kappa_y \frac{\partial B_y}{\partial t} + \frac{\sigma_y}{\varepsilon_0} B_y = \mu_1 \frac{\partial H_y}{\partial t}, \tag{43}$$

setting

$$\begin{array}{lll} ax = 2\kappa_x / \Delta t + \sigma_x / 2\varepsilon_0 & ay = 2\kappa_y / \Delta t + \sigma_y / 2\varepsilon_0 & er = 2/(\varepsilon_1 \Delta t) \\ bx = 2\kappa_x / \Delta t - \sigma_x / 2\varepsilon_0 & by = 2\kappa_y / \Delta t - \sigma_y / 2\varepsilon_0 & hr = \Delta t / (2\mu_1) \end{array} \tag{44}$$

In sub-step $n \rightarrow n + 1/2$, discretize equation (38) as

$$D_z^{n+\frac{1}{2}}(i,j) = \frac{bx}{ax} \cdot D_z^n(i,j) + \frac{1}{ax} \cdot \left\{ \frac{1}{\Delta x} \cdot \left[ H_y^{n+\frac{1}{2}}\left(i+\frac{1}{2},j\right) - H_y^{n+\frac{1}{2}}\left(i-\frac{1}{2},j\right) \right] - \frac{1}{\Delta y} \cdot \left[ H_x^n\left(i,j+\frac{1}{2}\right) - H_x^n\left(i,j-\frac{1}{2}\right) \right] \right\}. \tag{45}$$

The discretization scheme of Equation (39) is

$$E_z^{n+\frac{1}{2}}(i,j) = \frac{by}{ay} \cdot E_z^n(i,j) + \frac{er}{ay} \cdot \left[ D_z^{n+\frac{1}{2}}(i,j) - D_z^n(i,j) \right], \tag{46}$$

where

$$D_z^{n+\frac{1}{2}}(i,j) = D_z^n(i,j) + \frac{ay}{er} \cdot E_z^{n+\frac{1}{2}}(i,j) - \frac{by}{er} \cdot E_z^n(i,j). \tag{47}$$

The discretization schemes of Equations (40)–(43), respectively, are

$$B_x^{n+\frac{1}{2}}\left(i,j+\frac{1}{2}\right) = \frac{by}{ay} \cdot B_x^n\left(i,j+\frac{1}{2}\right) - \frac{1}{ay} \cdot \frac{1}{\Delta y} \cdot [E_z^n(i,j+1) - E_z^n(i,j)], \tag{48}$$

$$B_y^{n+\frac{1}{2}}\left(i+\frac{1}{2},j\right) = \frac{bx}{ax} \cdot B_y^n\left(i+\frac{1}{2},j\right) + \frac{1}{ax} \cdot \frac{1}{\Delta x} \cdot \left[ E_z^{n+\frac{1}{2}}(i+1,j) - E_z^{n+\frac{1}{2}}(i,j) \right], \tag{49}$$

$$H_x^{n+\frac{1}{2}}\left(i,j+\frac{1}{2}\right) = H_x^n\left(i,j+\frac{1}{2}\right) + hr \cdot \left[ ax \cdot B_x^{n+\frac{1}{2}}\left(i,j+\frac{1}{2}\right) - bx \cdot B_x^n\left(i,j+\frac{1}{2}\right) \right], \tag{50}$$

$$H_y^{n+\frac{1}{2}}\left(i+\frac{1}{2},j\right) = H_y^n\left(i+\frac{1}{2},j\right) + hr \cdot \left[ ay \cdot B_y^{n+\frac{1}{2}}\left(i+\frac{1}{2},j\right) - by \cdot B_y^n\left(i+\frac{1}{2},j\right) \right], \tag{51}$$

In Equations (45)–(51), except for Equation (48), both sides contain the field component at time $(n + 1/2)$. By eliminating the simultaneous components from Equations (45)–(51), except for $E_z^{n+1/2}$, we obtain

$$a_i E_z^{n+\frac{1}{2}}(i-1,j) + b_i E_z^{n+\frac{1}{2}}(i,j) + c_i E_z^{n+\frac{1}{2}}(i+1,j) = d_i, \tag{52}$$

where

$$a_i = -\frac{hr}{(\Delta x)^2} \cdot \frac{ay}{ax} \quad b_i = \frac{ax \cdot ay}{er} + \frac{hr}{(\Delta x)^2} \cdot \frac{2 \cdot ay}{ax} \quad c_i = -\frac{hr}{(\Delta x)^2} \cdot \left[ ay\left(i + \tfrac{1}{2}, j\right) / ax\left(i + \tfrac{1}{2}, j\right) \right]$$

$$d_i = \frac{ax \cdot by}{er} E_z^n(i,j) + \left\{ \frac{1}{\Delta x} \cdot \left[ H_y^n\left(i + \tfrac{1}{2}, j\right) - H_y^n\left(i - \tfrac{1}{2}, j\right) \right] - \frac{1}{\Delta y} \cdot \left[ H_x^n\left(i, j + \tfrac{1}{2}\right) - H_x^n\left(i, j - \tfrac{1}{2}\right) \right] \right\}$$

$$+ (bx - ax) \cdot D_z^n(i,j) + \frac{hr}{\Delta x} \cdot \left[ \frac{ay \cdot bx}{ax} - by \right] \cdot B_y^n\left(i + \tfrac{1}{2}, j\right) - \left[ \frac{ay \cdot bx}{ax} - by \right] \cdot B_y^n\left(i - \tfrac{1}{2}, j\right). \tag{53}$$

In the sub-step of $n + 1/2 \rightarrow n + 1$, the difference scheme of Equation (38) is the same as above, which can be expressed as

$$D_z^{n+1}(i,j) = \frac{bx}{ax} \cdot D_z^{n+\frac{1}{2}}(i,j) + \frac{1}{ax} \cdot \left\{ \frac{1}{\Delta x} \cdot \left[ H_y^{n+\frac{1}{2}}\left(i + \tfrac{1}{2}, j\right) - H_y^{n+\frac{1}{2}}\left(i - \tfrac{1}{2}, j\right) \right] - \frac{1}{\Delta y} \cdot \left[ H_x^{n+1}\left(i, j + \tfrac{1}{2}\right) - H_x^{n+1}\left(i, j - \tfrac{1}{2}\right) \right] \right\}. \tag{54}$$

Equation (39) can be discretized as

$$E_z^{n+1}(i,j) = \frac{by}{ay} \cdot E_z^{n+\frac{1}{2}}(i,j) + \frac{er}{ay} \cdot \left[ D_z^{n+1}(i,j) - D_z^{n+\frac{1}{2}}(i,j) \right], \tag{55}$$

where

$$D_z^{n+1}(i,j) = D_z^{n+\frac{1}{2}}(i,j) + \frac{ay}{er} \cdot E_z^{n+1}(i,j) - \frac{by}{er} \cdot E_z^{n+\frac{1}{2}}(i,j). \tag{56}$$

Equations (40)–(43), respectively, can be discretized as

$$B_x^{n+1}\left(i, j + \tfrac{1}{2}\right) = \frac{by}{ay} \cdot B_x^{n+\frac{1}{2}}\left(i, j + \tfrac{1}{2}\right) - \frac{1}{ay} \cdot \frac{1}{\Delta y} \cdot \left[ E_z^{n+1}(i, j + 1) - E_z^{n+1}(i,j) \right], \tag{57}$$

$$B_y^{n+1}\left(i + \tfrac{1}{2}, j\right) = \frac{bx}{ax} \cdot B_y^{n+\frac{1}{2}}\left(i + \tfrac{1}{2}, j\right) + \frac{1}{ax} \cdot \frac{1}{\Delta x} \cdot \left[ E_z^{n+\frac{1}{2}}(i + 1, j) - E_z^{n+\frac{1}{2}}(i,j) \right], \tag{58}$$

$$H_x^{n+1}\left(i, j + \tfrac{1}{2}\right) = H_x^{n+\frac{1}{2}}\left(i, j + \tfrac{1}{2}\right) + hr \cdot \left[ ax \cdot B_x^{n+1}\left(i, j + \tfrac{1}{2}\right) - bx \cdot B_x^{n+\frac{1}{2}}\left(i, j + \tfrac{1}{2}\right) \right], \tag{59}$$

$$H_y^{n+1}\left(i + \tfrac{1}{2}, j\right) = H_y^{n+\frac{1}{2}}\left(i + \tfrac{1}{2}, j\right) + hr \cdot \left[ ay \cdot B_y^{n+1}\left(i + \tfrac{1}{2}, j\right) - by \cdot B_y^{n+\frac{1}{2}}\left(i + \tfrac{1}{2}, j\right) \right]. \tag{60}$$

Eliminating the simultaneous components of Equations (54)–(60), the time domain advance calculation is obtained as follows:

$$a_j E_z^{n+1}(i, j - 1) + b_j E_z^{n+1}(i,j) + c_j E_z^{n+1}(i, j + 1) = d_j, \tag{61}$$

where

$$a_i = -\frac{hr}{(\Delta y)^2} \cdot \frac{ax}{ay} \quad b_i = \frac{ax \cdot ay}{er} + \frac{hr}{(\Delta y)^2} \cdot \frac{2ax}{ay} \quad c_i = -\frac{hr}{(\Delta y)^2} \cdot \frac{ax}{ay}$$

$$d_i = \frac{ax \cdot by}{er} E_z^{n+\frac{1}{2}}(i,j) + \left\{ \frac{1}{\Delta x} \cdot \left[ H_y^{n+\frac{1}{2}}\left(i + \tfrac{1}{2}, j\right) - H_y^{n+\frac{1}{2}}\left(i - \tfrac{1}{2}, j\right) \right] - \frac{1}{\Delta y} \cdot \left[ H_x^{n+\frac{1}{2}}\left(i, j + \tfrac{1}{2}\right) - H_x^{n+\frac{1}{2}}\left(i, j - \tfrac{1}{2}\right) \right] \right\}$$

$$+ (bx - ax) \cdot D_z^{n+\frac{1}{2}}(i,j) - \frac{hr}{\Delta y} \cdot \left[ \left( \frac{ax \cdot by}{ay} - bx \right) \cdot B_x^{n+\frac{1}{2}}\left(i, j + \tfrac{1}{2}\right) - \left( \frac{ax \cdot by}{ay} - bx \right) \cdot B_x^{n+\frac{1}{2}}\left(i, j - \tfrac{1}{2}\right) \right]. \tag{62}$$

The steps of ADI-FDTD for the UPML boundary of 2D TM waves are as follows:

In the time step of $n \rightarrow n + 1/2$:

(1) Calculate $E_z^{n+1/2}$ by Equation (52);
(2) Calculate $D_z^{n+1/2}$ by Equation (47);
(3) Calculate $B_x^{n+1/2}$ and $B_y^{n+1/2}$ by Equations (48) and (49);
(4) Calculate $H_x^{n+1/2}$ and $H_y^{n+1/2}$ by Equations (50) and (51).

In the time step of $n+1/2 \rightarrow n+1$:

(1) Calculate $E_z^{n+1}$ by Equation (61);

(2) Calculate $D_z^{n+1}$ by Equation (56),

(3) Calculate $B_x^{n+1}$ and $B_y^{n+1}$ by Equations (57) and (58);

(4) Calculate $H_x^{n+1}$ and $H_y^{n+1}$ by Equations (59) and (60).

We used a simple air model to verify the absorption effect of the UPML boundary conditions. Visual Studio 2010 was used as a development tool, and the CPU was an Intel Core i5-4200H with an NVIDIA GeForce GTX 950m. The computing environment above supported all the computing processes in this paper. The simulation area was a 2.0 m × 2.0 m rectangular area; the spatial step was 0.005 m, and the time step was $\Delta t$ = 0.01 ns. A Ricker wavelet was added at the positive center of the model's area (see Figure 3). Figure 4 shows the snapshots of the $E_z$ field at different moments. It can be seen that the UPML absorption boundary condition had no obvious reflections, and the absorption effect was good.

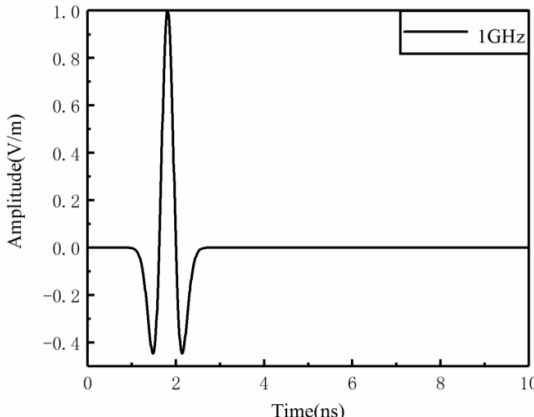

**Figure 3.** The waveform of the Ricker wavelet.

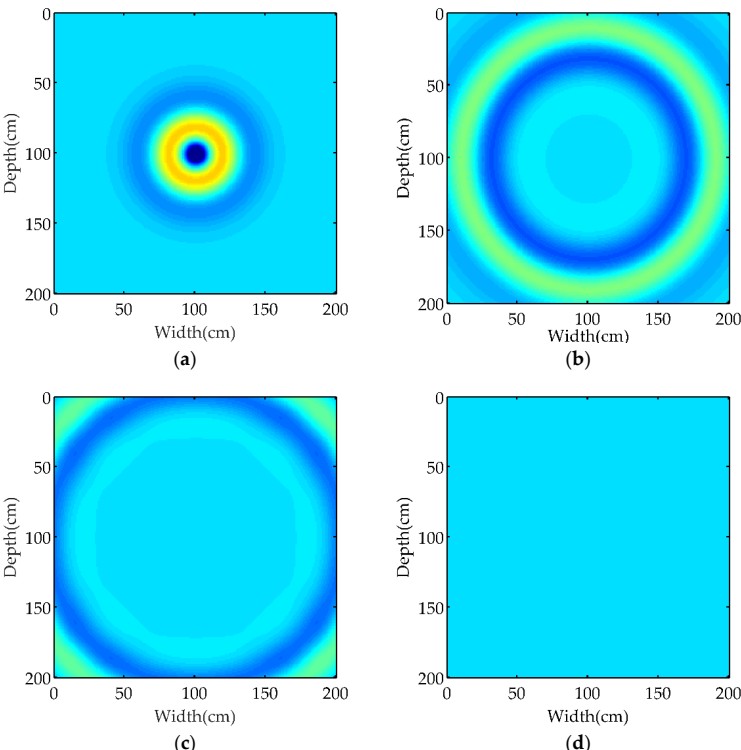

**Figure 4.** Snapshots of the wave field of $E_z$ at different times: (**a**) At 2 ns; (**b**) At 3 ns; (**c**) At 4 ns; (**d**) At 5 ns.

## 3. Numerical Simulations

### 3.1. Two-Layer Medium Model with Circular Cavity

As shown in Figure 5, the model area was set to 2.0 m × 2.0 m. The upper layer of the model was a 0.2 m air layer, and the lower layer was a 1.8 m clay layer. The main frequency of the excitation source was a 1 GHz Ricker wavelet, which was located at a 0.2 m depth in the model, and the transmitter and receiver were 0.1 m apart. The relative dielectric constant of the clay layer was 12, and the conductivity was 2 mS/m. A circular cavity with a diameter of 0.1 m was set in the clay layer. The relative dielectric constant and conductivity were 30 and 0 mS/m, respectively. The relative magnetic conductivity of all the material was 1. We set the spatial step size to 0.005 m, the time step size to 0.01 ns, and the thickness of the UPML to 0.01 m.

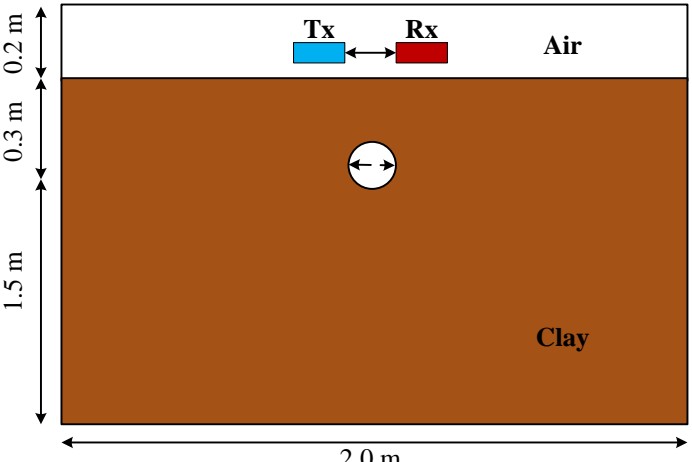

**Figure 5.** Schematic diagram of the circular cavity model.

The time step of the conventional FDTD algorithm needs to satisfy the CFL stability condition. However, the ADI-FDTD algorithm is unconditionally stable and can save computation time by choosing a larger time step. Figure 6 shows a comparison of the $E_z$ field distributions of the conformal ADI-FDTD with different CFLN, the non-conformal FDTD, and the non-conformal ADI-FDTD. The CFLN is the ratio of the time step of the ADI-FDTD algorithm to that of the FDTD algorithm. It can be seen that the results of the non-conformal FDTD, the non-conformal ADI FDTD algorithm, and the conformal ADI FDTD algorithm have good consistency in different time steps. Table 1 provides the calculation time of the different algorithms when simulating the circular cavity model, respectively. When CFLN is 5, compared with the non-conformal FDTD algorithm, the time step of the conformal ADI-FDTD algorithm is increased to 5 times, and the number of iterations is reduced to 1/5. At this point, the computation time of the conformal ADI-FDTD algorithm is 961 s and that of the non-conformal FDTD algorithm is 1849 s, saving about 48% of the time. Figure 7 shows the GPR profiles obtained by the non-conformal FDTD method, non-conformal ADI-FDTD method, and the conformal ADI-FDTD method. We indicate multiple reflections within the cavity with red boxes. As can be seen from Figure 7, the images of the non-conformal FDTD method and the non-conformal ADI-FDTD method are consistent. The conformal ADI-FDTD method has fewer multiple reflections than the non-conformal ADI-FDTD method. At the same time, the conformal ADI-FDTD method is closer to the conformal FDTD method than the non-conformal ADI-FDTD method. This indicates that the conformal ADI-FDTD method can effectively reduce the false reflection waves caused by conventional ladder approximation when simulating an underground circular cavity. Table 1 and Figure 7 show that the conformal ADI-FDTD algorithm can greatly improve the computational efficiency and can also effectively improve the computational accuracy.

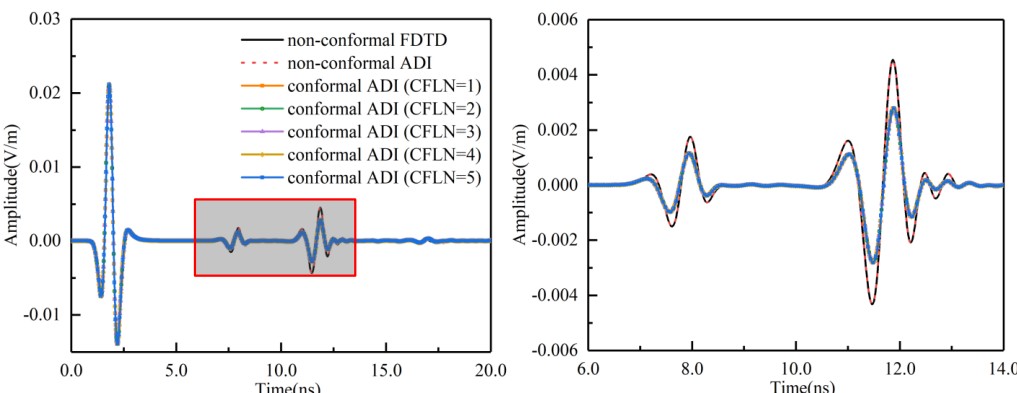

**Figure 6.** Comparison of $E_z$ field distribution between ADI−FDTD and FDTD simulation.

**Table 1.** Computational resource usage of different methods.

| Method | Memory (Mb) | $\Delta t$ (ns) | Iterations | Time (s) |
|---|---|---|---|---|
| Non-conformal FDTD | 10.16 | 0.01 | 5000 | 1849 |
| Conformal FDTD | 10.20 | 0.01 | 5000 | 1897 |
| Non-conformal ADI-FDTD | 22.39 | 0.01 | 5000 | 4765 |
| Conformal ADI-FDTD-CFLN = 1 | 22.48 | 0.01 | 5000 | 4812 |
| Conformal ADI-FDTD-CFLN = 2 | 22.48 | 0.02 | 2500 | 2403 |
| Conformal ADI-FDTD-CFLN = 3 | 22.48 | 0.03 | 1667 | 1608 |
| Conformal ADI-FDTD-CFLN = 4 | 22.48 | 0.04 | 1250 | 1196 |
| Conformal ADI-FDTD-CFLN = 5 | 22.48 | 0.05 | 1000 | 961 |

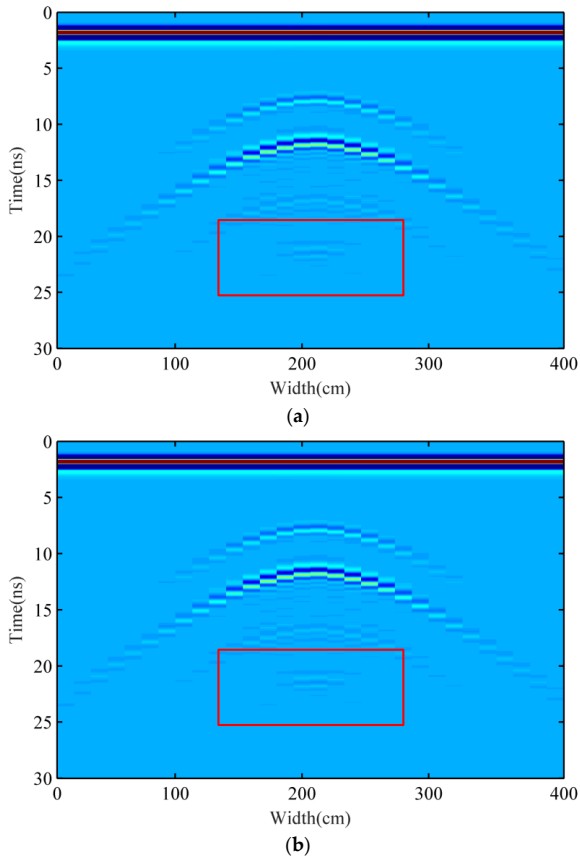

(a)

(b)

**Figure 7.** *Cont.*

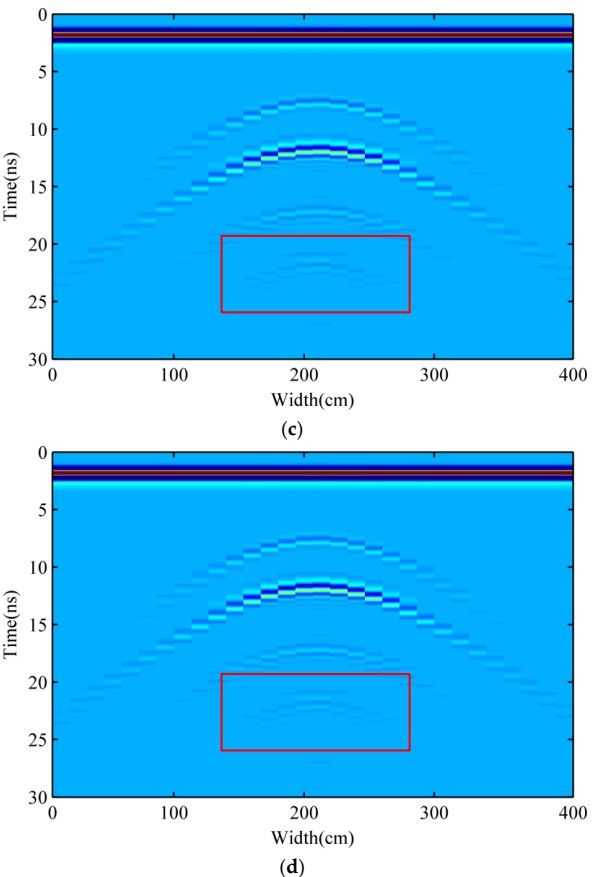

(c)

(d)

**Figure 7.** GPR B-scan images of the two-layer cavity model: (**a**) Obtained by conformal ADI-FDTD method; (**b**) Obtained by conformal FDTD method; (**c**) Obtained by non-conformal ADI-FDTD method; (**d**) Obtained by non-conformal FDTD method.

### 3.2. Single-Pipe Model with Cavity Disease

This model simulated two types of the cavity disease around the pipeline, which further verifies the accuracy of the conformal algorithm. The model had two layers with an air layer on top and a clay layer on the bottom. As shown in Figure 8, there were some irregular cavities above the concrete pipe with an outer diameter of 0.6 m and an inner diameter of 0.48 m. Figure 9 shows a concrete pipe with an outer diameter of 1.2 m and an inner diameter of 0.96 m with some irregular cavities underneath. The spatial steps and time steps were 0.005 m and 0.01 ns, respectively, and the simulation iterations were 5000. Table 2 shows the conductivity and relative permittivity of different materials in the model. The excitation source of the model was the same as above.

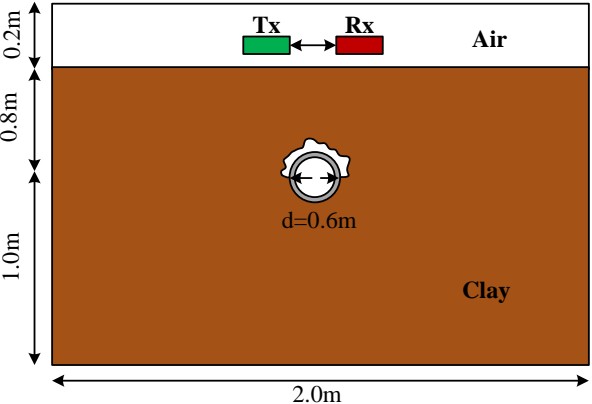

**Figure 8.** Schematic diagram of model with cavity disease on the upper part of pipeline.

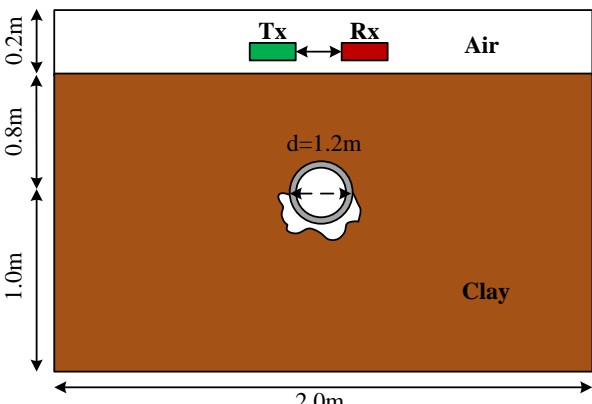

**Figure 9.** Schematic diagram of model with cavity disease on the lower part of pipeline.

**Table 2.** Conductivity and dielectric constant of different media.

| Material | $\varepsilon_r$ | $\sigma$ (S/m) |
|---|---|---|
| Air | 1 | 0 |
| Clay | 12 | 0.002 |
| PVC plastic | 3 | 0 |
| Concrete | 6 | 0.001 |
| Metal | 1 | $1.0 \times 10^6$ |

By analyzing Figure 10, the non-conformal model profile only shows one distinct reflected wave at the bottom of the pipe. The profile of the conformal model shows that there are two reflected waves on the upper and lower sides of the pipe, and the reflected waves of the irregular cavity disease above the pipe are incredibly apparent. As shown in Figure 11, both the non-conformal and conformal model profiles clearly show the reflected waves at the top and bottom of the pipe, but the non-conformal model profile only shows the reflected waves from the cavity damage on the left side, while the conformal model profile completely shows the reflected waves distributed on both sides of the cavity damage. As can be seen in Figures 10 and 11, the circular pipe grid points are more accurately processed using the conformal grid technique, and the conformal ADI FDTD algorithm is more accurate.

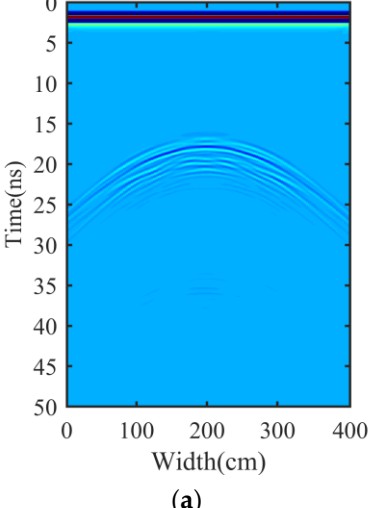

(**a**)

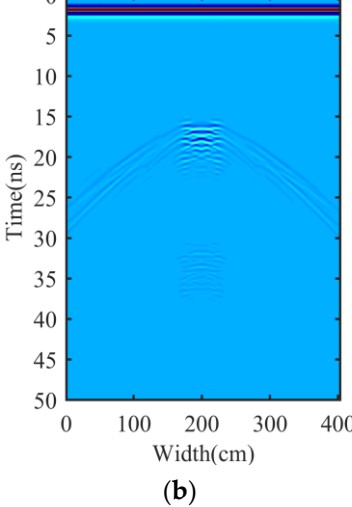

(**b**)

**Figure 10.** GPR B-scan image of a disease model with an irregular cavity above pipeline: (**a**) Obtained by non-conformal method; (**b**) Obtained by the conformal method.

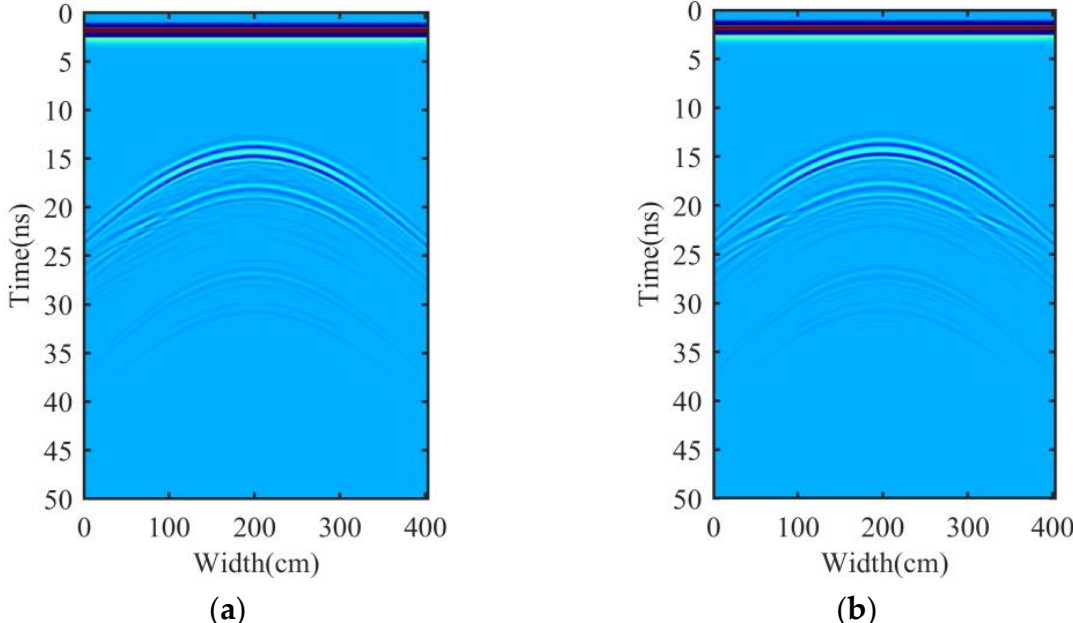

**(a)**　　　　　　　　　　　　　　**(b)**

**Figure 11.** GPR B-scan image of a disease model with an irregular cavity below pipeline: (**a**) Obtained by non-conformal method; (**b**) Obtained by the conformal method.

### 3.3. Complex Multi-Pipe Model

This model used the conformal ADI-FDTD algorithm to simulate the complex multi-tube model. Figures 12 and 13 show the schematic diagrams of complex multi-pipe models. Figure 12 shows an underground concrete multi-pipe model, with pipe diameters of 0.03 m, 0.06 m, and 0.1 m. Figure 13 shows an underground multi-pipe model with pipe diameters of 0.4 m. The pipe materials were concrete, metal, and PVC plastic. The space step and time step were set as 0.005 m and 0.01 ns, and simulation iterations were 5000. The model had two layers: the upper layer was an air layer, and the lower layer was a clay layer. Table 2 provides the relative permittivity and conductivity. The excitation source and GPR system were the same as for the single-pipe model.

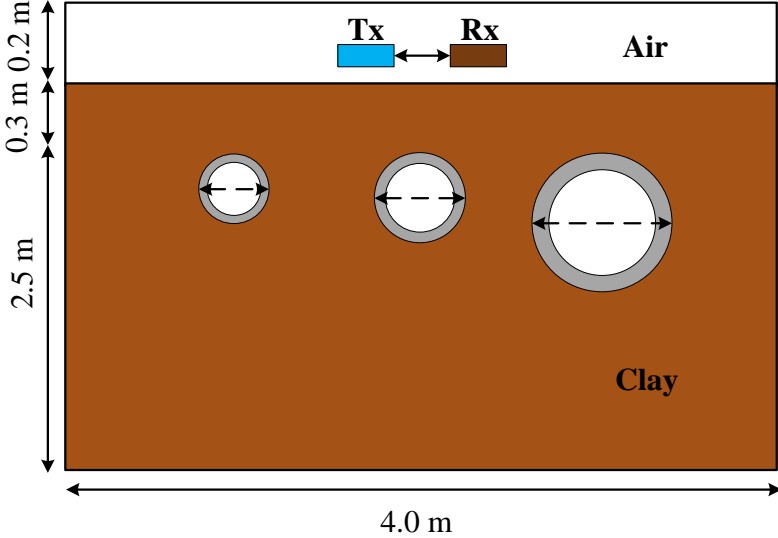

**Figure 12.** Underground multi-pipe model with different pipe diameters.

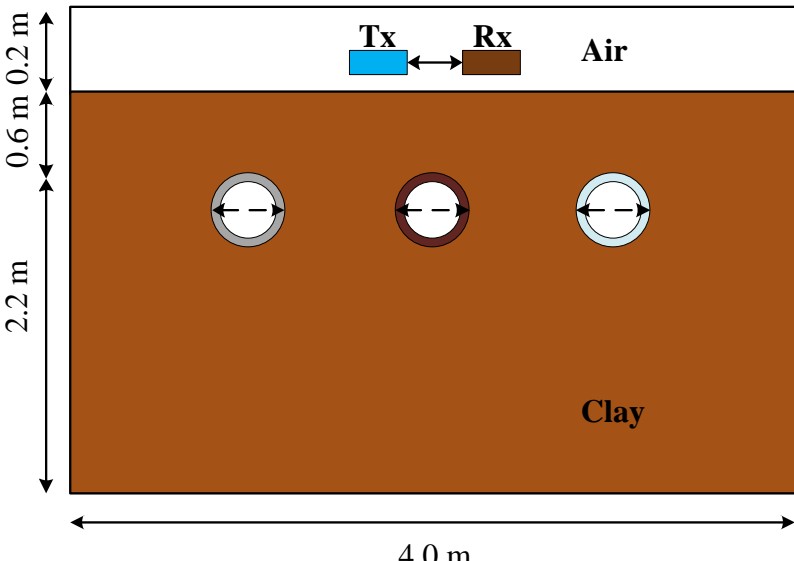

**Figure 13.** Underground multi-pipe model with different pipe materials.

Figures 14 and 15 show the GPR B-scan images obtained by the underground multipipe model using the conformal ADI-FDTD algorithm. Figure 14, G1, G2, and G3 indicate the intersection points of the hyperbolic asymptotes at the tops of concrete pipes with diameters of 0.3 m, 0.6 m, and 1.0 m, respectively. Reflected waves are evident at the top and bottom of the pipeline; the more reflective the diameter, the more pronounced the reflected waves are. There are multiple reflected and stray waves at the bottom of the profile image. In Figure 15, the PVC plastic pipe on the right side of the profile has weak reflected waves, and the concrete pipeline located on the left side has distinct hyperbolic reflected waves. The reflected waves are evident at the top of the middle metal tube, while not at the bottom. In addition, some interfering stray waves and multiple reflected waves can be observed at the bottom of the image.

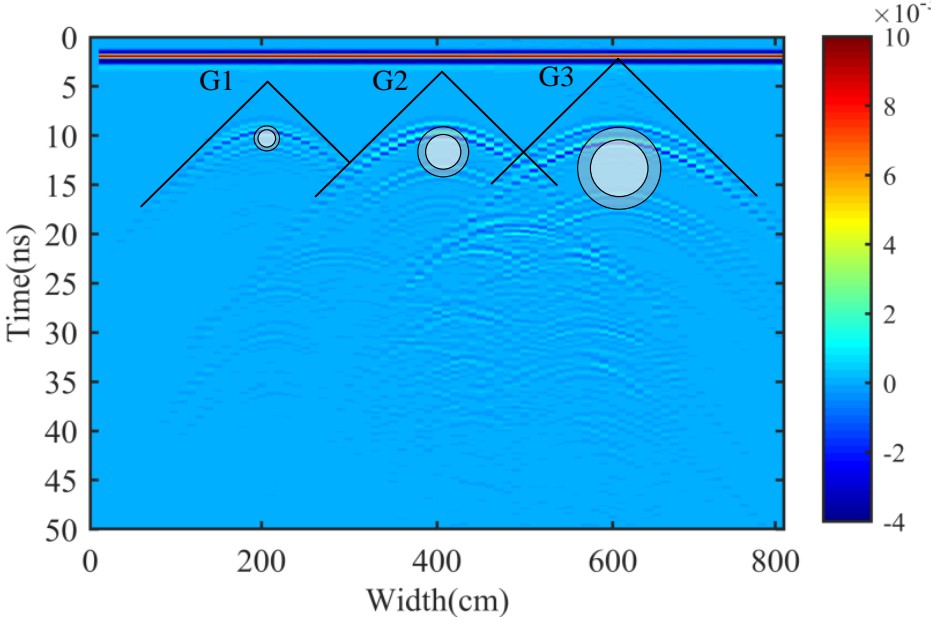

**Figure 14.** Simulation results of underground multi−pipe model with different pipe diameters.

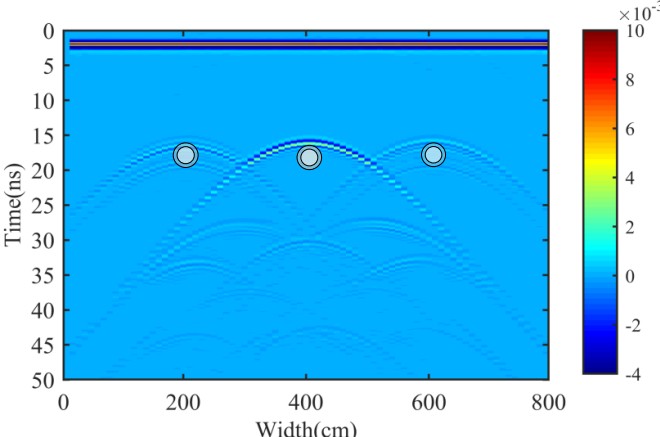

**Figure 15.** Simulation results of underground multi−pipe model with different pipe materials.

As shown in Figures 14 and 15, for an underground multi-pipe model with the same depth and materials but different diameters, an increase in diameter causes the intersection position of the hyperbolic asymptote at the top of the pipe to become higher, and the reflection waves at the bottom and top of the pipe will be more obvious. As the thickness of the pipe wall is related to the pipe's diameter, two obvious hyperbolas with larger diameters can be seen at the bottom and top of the concrete pipe. For underground multi-pipe models with the same diameter and composed of different materials, the closer the dielectric properties of the pipe material are to those of the surrounding clay, the weaker its response to the electromagnetic waves and the less pronounced the diffraction hyperbolic features are at the top and bottom. Due to the strong response of electromagnetic waves to metals, the diffraction hyperbolic characteristics at the top of the metal pipe were very obvious, while there were no hyperbolic characteristics at the bottom of the pipe and fewer multiple reflections occurred inside.

### 3.4. Complex Underground Structures

We used the conformal ADI-FDTD algorithm to simulate complex underground structure models. Figure 16 shows schematic diagrams of rectangular and circular void models. There was a metal pipe located 0.5 m below the surface with an outside diameter of 0.3 m and an inside diameter of 0.2 m. There was a rectangular cavity of a size of 0.1 m on the left side of the metal pipe. On the other side, there was a circular cavity of a diameter of 0.1 m. The space step and time step were set as 0.005 m and 0.01 ns, and simulation iterations were 5000. The model had two layers: the upper layer was an air layer, and the lower layer was a clay layer. Table 2 provides the relative permittivity and conductivity. The excitation source and GPR system were the same as for the previous model.

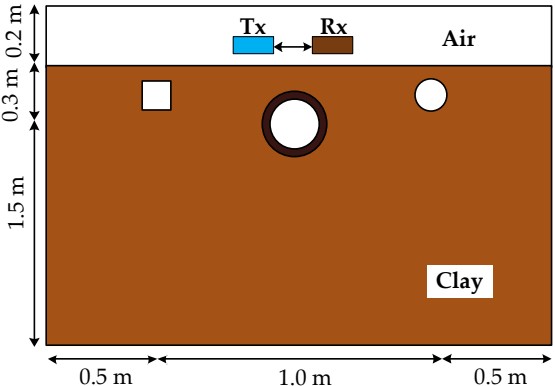

**Figure 16.** Schematic diagram of complex underground cavity model.

Figure 17 shows the GPR B-scan image obtained by the underground cavity model using the conformal ADI-FDTD algorithm. By analyzing Figure 17, the metal pipe produces strong reflected and diffracted waves. It is due to the strong response of the electromagnetic waves to metals. The upper interface of the rectangular cavity is still a flat interface in the radar forward image. Meanwhile, the lower interface is similar to the upper interface but retains weak reflected wave energy. The circular cavity produces a clear reflected wave, and the diffracted wave is weaker due to its smaller size. Moreover, some mutual interference clutters and multiple reflection waves can be observed at the bottom of the image. By analyzing the GPR scans, the characteristics of the model can be inferred more comprehensively and accurately, helping to interpret the GPR information.

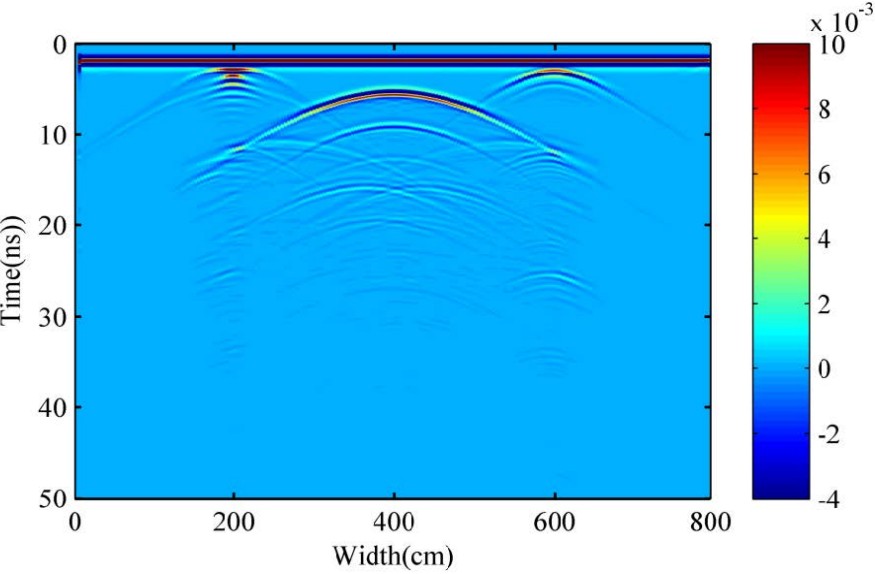

**Figure 17.** GPR B−scan images of underground cavity mode.

## 4. Field Experiment

The correctness and effectiveness of the conformal ADI-FDTD algorithm were verified by a GPR detection experiment of rectangular voids. As shown in Figure 18, a carton with a length of 0.5 m, a width of 0.3 m, and a height of 0.16 m was buried in the soil, and the top surface of the carton was 0.16m from the ground. In this experiment, GSSI SIR-4000 road radar was used for detection, and the center frequency was 400 MHz.

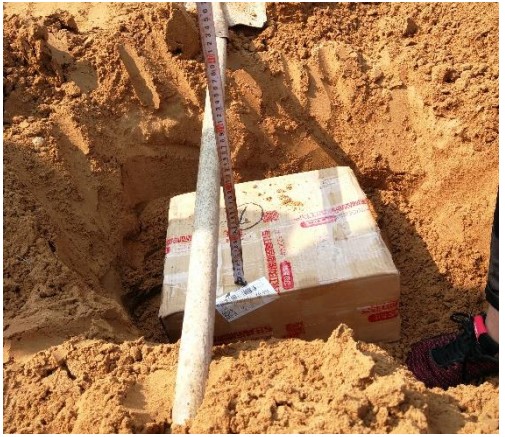 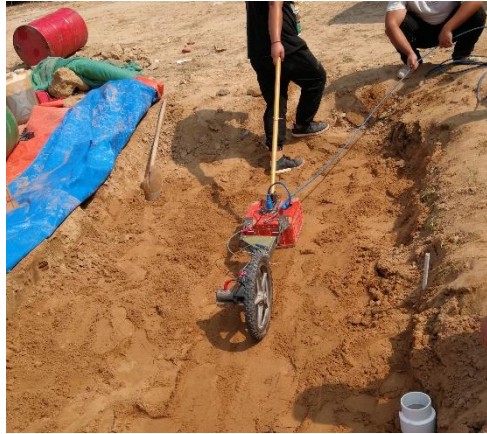

**Figure 18.** Road radar rectangular cavity experimental diagram.

The experimental model was simplified to a two-dimensional case, as shown in Figure 19. It was assumed that the relative permittivity of the soil layer was 6.8 and the

conductivity was 3 mS/m. The time step was set to 0.001 ns; the space step was set to 0.005 m, and the simulation iterations were taken as 2000. Figure 20 was a single-channel wave comparison diagram of the numerical simulation results of the conformal ADI-FDTD algorithm and the measured results.

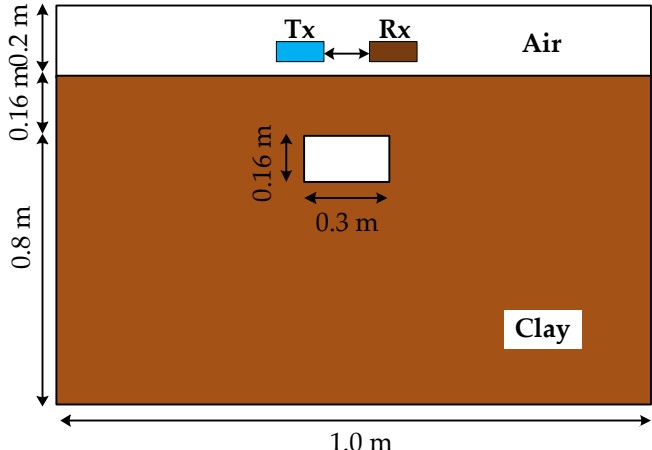

**Figure 19.** Two-dimensional experimental model.

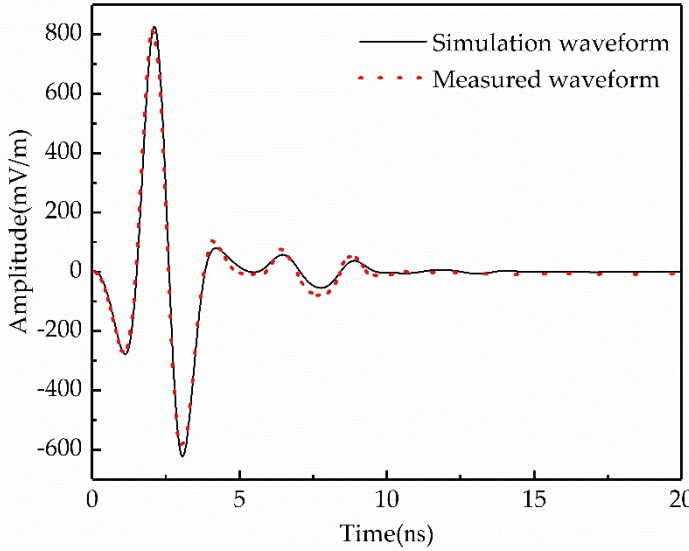

**Figure 20.** Single−channel wave comparison diagram of simulated and measured results.

It can be seen from Figure 20 that the simulated waveform obtained by the algorithm in this paper is in good agreement with the measured waveform in terms of amplitude and delay, with differences only in details. This is mainly due to the fact that we considered the soil layer as a homogeneous and isotropic medium, which led to the difference between the assumed dielectric constant of the material and the real dielectric constant of the material during the simulation. This example verifies the applicability and effectiveness of the proposed algorithm for practical engineering inspection.

## 5. Conclusions

In this study, an efficient and accurate GPR forward model based on the ADI-FDTD method and surface conformal technology was established. The GPR image characteristics of the different shaped cavity diseases and multiple underground pipeline structures with different materials and diameters were obtained. The model provides a basis for further processing and interpretation of the measured data of GPR electromagnetic waves. The numerical simulation results demonstrate that the conformal ADI-FDTD method can greatly

reduce the errors caused by the ladder approximation method, which is traditionally employed to divide a circular pipe grid. Moreover, when the time step is increased to 5 times, the conformal ADI-FDTD algorithm can save 48% of the computation time compared to the traditional FDTD method. The correctness and effectiveness of the conformal ADI-FDTD algorithm was verified by a field experiment. In future research, we intend to conduct GPR inversion analyses of the underground pipe structures. Through a 2D GPR image inversion analysis of the underground pipe structures, we will determine the positions and sizes of cavity diseases in underground pipes.

**Author Contributions:** Methodology, Y.L. and F.W.; software, Y.L. and N.W.; validation, J.L.; writing—original draft preparation, Y.L.; writing—review and editing, J.L. and N.W.; supervision, F.W. and C.L.; investigation, C.L. All authors have read and agreed to the published version of the manuscript.

**Funding:** This research was funded by [the National Key Research and Development Program of China] grant number [2017YFC1501204], [the National Natural Science Foundation of China] grant number [11771407], [the Program for Science and Technology Innovation Talents in Universities of Henan Province] grant number [19HASTIT043], and [the Outstanding Young Talent Research Fund of Zhengzhou University] grant number [1621323001]. And The APC was funded by [the Program for Innovative Research Team at the University of Henan Province] grant number [18IRTSTHN007].

**Institutional Review Board Statement:** Not applicable.

**Informed Consent Statement:** Not applicable.

**Data Availability Statement:** Not applicable.

**Acknowledgments:** This work was supported by the National Key Research and Development Program of China (No. 2017YFC1501204), the National Natural Science Foundation of China (No. 11771407), the Program for Science and Technology Innovation Talents in Universities of Henan Province (No. 19HASTIT043), the Outstanding Young Talent Research Fund of Zhengzhou University (1621323001), and the Program for Innovative Research Team at the University of Henan Province (18IRTSTHN007). The authors would like to thank for these financial supports.

**Conflicts of Interest:** The authors declare no conflict of interest.

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
