# Peer review of "Modeling GPR Wave Propagation in Complex Underground Structures Using Conformal ADI-FDTD Algorithm"

_applsci, doi:10.3390/app12105219_

Round 1

Reviewer 1 Report

The authors described efficient and accurate GPR forward model based on the ADI-FDTD method.  Methodology and numerical simulations are clearly described. The authors obtained The GPR image characteristics of the different shapes cavity diseases and multiple underground pipeline structures with different materials and diameters. The numerical simulation results demonstrated that the conformal ADI-FDTD method can greatly reduce the errors caused by the ladder approximation method. I recommend to publish this paper at present form.

Author Response

Thank you very much for the reviewer's affirmation of this manuscript.

Reviewer 2 Report

This paper developed a GPR model by using ADI-FDTD method and verified the algorithm with the field experiment. Overall the paper was well written and the results were scientifically sounded. The positive note about this paper was that there were comparisons among the traditional FDTD and ADI-FDTD algorithms.

However, the authors should provide supporting evidence regarding the computational resources used for each algorithm and explain why the ADI-FDTD could reduce up to 48% computational time.

Author Response

  1. The authors should provide supporting evidence regarding the computational resources used for each algorithm.

Author's Response:

We appreciate the reviewer’s comment. We have added the computational environment for all computational processes in Section 2.3, page 9, line 210, and provide supporting evidence for the computational resources used by each algorithm in Table 1, as follows:

“We used a simple air model to verify the absorption effect of UPML boundary conditions. Visual Studio 2010 was used as a development tool, and the CPU was an Intel Core i5-4200H with an NVIDIA GeForce GTX 950m. The computing environment above supported all the computing processes in this paper. The simulation area was a 2.0 m × 2.0 m rectangular area, the spatial step was 0.005 m, and the time step was Δt = 0.01 ns.”

Table 1. Computational resource usage of different methods.

Method

Memory (Mb)

Δt (ns)

Iterations

Time (s)

Non-conformal FDTD

10.16

0.01

5000

1849

Conformal FDTD

10.20

0.01

5000

1897

Non-conformal ADI-FDTD

22.39

0.01

5000

4765

Conformal ADI-FDTD-CFLN=1

22.48

0.01

5000

4812

Conformal ADI-FDTD-CFLN=2

22.48

0.02

2500

2403

Conformal ADI-FDTD-CFLN=3

22.48

0.03

1667

1608

Conformal ADI-FDTD-CFLN=4

22.48

0.04

1250

1196

Conformal ADI-FDTD-CFLN=5

22.48

0.05

1000

961

  1. The authors should explain why the ADI-FDTD could reduce up to 48% computational time.

Author's Response:

We agree with the referee’s recommendation, and modified the description in section 3.1 (in paragraph 1 page 12), as follows:

The time step of the conventional FDTD algorithm needs to satisfy the CFL stability condition. However, the ADI-FDTD algorithm is unconditionally stable and can save computation time by choosing a larger time step. Figure 6 shows a comparison of the Ez field distributions of the conformal ADI-FDTD with different CFLN, the non-conformal FDTD and the non-conformal ADI-FDTD. The CFLN is the ratio of the time step of the ADI-FDTD algorithm to that of the FDTD algorithm. It can be seen that the results of the non-conformal FDTD, the non-conformal ADI FDTD algorithm and the conformal ADI FDTD algorithm have good consistency in different time steps. Table 1 provides the calculation time of the different algorithms when simulating the circular cavity model, respectively. When CFLN is 5, compared with the non-conformal FDTD algorithm, the time step of the conformal ADI-FDTD algorithm is increased to 5 times, and the number of iterations is reduced to 1/5. At this point, the computation time of the conformal ADI-FDTD algorithm is 961 s and that of the non-conformal FDTD algorithm is 1849 s, saving about 48% of the time.

Reviewer 3 Report

In this paper, an efficient and accurate GPR forward model based on the ADI-FDTD method and surface conformal technology was established. The GPR image characteristics of the different shapes of cavity diseases and multiple underground pipeline structures with different materials and diameters were obtained. A field experiment proves the correctness and effectiveness of the proposed algorithm in actual detection.
The research design is appropriate. The methods and results are adequately described. The results are clearly presented. The conclusions are supported by the results.

The Introduction should be expanded and reflect current research on the topic of the article. Of the 42 references, only 4 are dated after 2015. Therefore, the Introduction must be redone.

In the introduction, the authors in some lines abuse citations (see line 66) 11 citations for a sentence! This is not acceptable unless the authors give full credit to each reference or reduce them to a reasonable number. Please check this issue elsewhere in the article.
There is no citation for [42] in the text.

Significant plagiarism is found in the abstract. Please check this issue!
Plagiarisms are found in 3.2 in the text without citation on pages 12 and 13. Please check this issue!

The Eqs. must be numbered in lines 100, 107, 119, 160, 166, 183.

Please check the paper for English editing and typos.

In general, the article makes a good impression, is devoted to an interesting and topical problem of simulating GPR electromagnetic wave propagation in pipe structures.

Author Response

1.The Introduction should be expanded and reflect current research on the topic of the article. Of the 42 references, only 4 are dated after 2015. Therefore, the Introduction must be redone.

Author's Response:

We appreciate the reviewer’s comment. Based on the reviewers' comments, we have expanded the Introduction section to reflect current research on the subject of this paper. At the same time, we added references dated after 2015. The specific correction is in line 46 of the second page, as follows:

Information about the location, size, and dielectric properties of complex underground structures can be obtained through inversion analysis of GPR echo signals [10-13]. Through accurate and efficient GPR forward modeling of complex underground structure models, the propagation law of GPR electromagnetic waves in underground structures can be obtained [14-16]. Which lays the theoretical foundation for inversion analysis and helps to improve the interpretation and processing accuracy of GPR measured data. At present, the most commonly used GPR forward simulation techniques include the finite element method (FEM) [17,18], the ray tracing method (RTM) [19,20], the method of moment (MOM) [21], the finite difference time domain method (FDTD) [22-24], the pseudo-spectral time domain method (PSTD) [25], and the symplectic algorithm [26-28], among others. Although research into GPR forward simulation has achieved fruitful results, these algorithms still have some limitations in terms of computational accuracy and efficiency. For example, the FEM may appear "pseudo-solving" phenomenon during calculation; the RTM cannot consider the dynamic characteristics of the electromagnetic wave of the ground-penetrating radar; while the time step of the symplectic algorithm needs to satisfy the Courant-Friedrichs-Lewy (CFL) stability condition, and the computational efficiency is limited. Therefore, it is necessary to propose a GPR forward modeling method with high computational accuracy and fast computational efficiency.

10. Keskinen J , Looms M C , Klotzsche A , et al. Practical data acquisition strategy for time-lapse experiments using crosshole GPR and full-waveform inversion. Journal of Applied Geophysics, 2021, 191(11):104362.

11. Kaplanvural S , Peken E , Zkap K . 1D waveform inversion of GPR trace by particle swarm optimization. Journal of Applied Geophysics, 2020, 181:104157.

12. Zhang F , Liu B , Wang J , et al. Two-dimensional Time-domain Full Waveform Inversion of On-ground Common-offset GPR Data Based on Integral Preprocessing. Journal of Environmental & Engineering Geophysics, 2020, 25(3):369-380.

13. Motevalli Z , Zakeri B . Time-Domain Spectral Inversion Method for Characterization of Subsurface Layers in Ground-Penetrating-Radar (GPR) Applications. Applied Computational Electromagnetics Society journal, 2019, 34(1):93-99.

15. Caselle C , Bonetto S , Comina C , et al. GPR surveys for the prevention of karst risk in underground gypsum quarries. Tunnelling and Underground Space Technology, 2020, 95:103137.

16. Park B , Kim J , Lee J , et al. Underground Object Classification for Urban Roads Using Instantaneous Phase Analysis of Ground-Penetrating Radar (GPR) Data. Remote Sensing, 2018, 10(9)

17. Xu X , Brekke C , Doulgeris A P , et al. Numerical Analysis of Microwave Scattering from Layered Sea Ice Based on the Finite Element Method. Remote Sensing, 2018, 10(9):1332.

18. Zhang W Y , Hao T , Chang Y , et al. Time-frequency analysis of enhanced GPR detection of RF tagged buried plastic pipes. NDT & E International, 2017, 92:88-96.

19. T. Langan. Seismic tomography: The accurate and efficient tracing of rays through heterogeneous media. Seg Technical Program Expanded Abstracts 1984, 3(1), 856.

23. Li Y , Zhao Z , Xu W , et al. An effective FDTD model for GPR to detect the material of hard objects buried in tillage soil layer. Soil and Tillage Research, 2019, 195:104353.

24. Alsharahi G , Faize A , Mostapha A , et al. 2D FDTD Simulation to Study Response of GPR Signals in Homogeneous and Inhomogeneous Mediums. International Journal on Communications Antenna and Propagation, 2016, 6(3):153.

25. Os A , Jlv A , Hv B . Two-step perfectly matched layer for arbitrary-order pseudo-spectral analytical time-domain methods. Computer Physics Communications, 2019, 235:102-110.

26. Yang M , Fang H , Wang F , et al. The Three Dimension First-Order Symplectic Partitioned Runge-Kutta Scheme Simulation for GPR Wave Propagation in Pavement Structure. IEEE Access, 2019, PP(99):1-1.

27. Fang H , Gao L . Symplectic partitioned Runge-Kutta methods for two-dimensional numerical model of ground penetrating radar. Computers & Geosciences, 2012, 49:323-329.

28. Lei J , Wang Z , Fang H , et al. Analysis of GPR Wave Propagation in Complex Underground Structures Using CUDA-Implemented Conformal FDTD Method. International Journal of Antennas and Propagation, 2019, 2019(11):1-11.

2. In the introduction, the authors in some lines abuse citations (see line 66) 11 citations for a sentence! This is not acceptable unless the authors give full credit to each reference or reduce them to a reasonable number. Please check this issue elsewhere in the article.

Author's Response:

We appreciate the reviewer’s comment. We reduced the citation in line 66 to a reasonable number and checked for this issue elsewhere in the article. As follow:

“The ADI-FDTD algorithm overcomes the CFL stability condition, and can employ a larger time step to improve the computing simulation efficiency [29-31].”

3. There is no citation for [42] in the text.

Author's Response:

We apologize for we made some mistakes, and we cite this literature on page 7, line 169.

“In this paper, the uniaxial anisotropic absorption layer (UPML) with easy programming and simple iterative formula is used as the absorption boundary [38].”

4. Significant plagiarism is found in the abstract. Please check this issue!

Author's Response:

We appreciate the reviewer’s comment, and the Abstract has been revised accordingly to reduce duplication rates, as follows:

Ground Penetrating Radar (GPR) is a shallow geophysical method for detecting and locating subsurface targets. The GPR image echo characteristics of complex underground structures can be obtained by carrying out GPR forward modeling research. The traditional finite-difference time-domain (FDTD) method has low efficiency and accuracy. The alternating direction implicit FDTD (ADI-FDTD) algorithm surmount the stability limitations of the traditional FDTD method, making it possible to select a larger time step for higher computational efficiency. For circular underground structures, pseudo wave produced by the ladder approximation method can be corrected using the surface conformal technique. This paper proposes a high-efficiency and high-accuracy GPR forward modeling method that combines ADI-FDTD algorithm and surface conformal technology. The performance of the conformal ADI-FDTD algorithm is verified by a simple two-layer model. Based on the proposed algorithm, the GPR image features of three complex underground structure models are obtained. Finally, a field experiment is used to support the accuracy and usefulness of the conformal ADI-FDTD algorithm. The numerical simulation results and experimental results exhibit that the conformal ADI-FDTD algorithm reduces the pseudo-diffraction wave caused by the ladder approximation method, and can significantly improve the computing efficiency for complex underground structure models.”

5. Plagiarisms are found in 3.2 in the text without citation on pages 12 and 13. Please check this issue!

Author's Response:

We appreciate the reviewer’s comment, and have made corresponding revisions to pages 12 and 13 in 3.2 to reduce the duplication rate, as follows:

“This model simulated two types of the cavity disease around pipeline, which further verifies the accuracy of the conformal algorithm. The model had two layers with an air layer on top and a clay layer on the bottom. As shown in Figure 8, there were some irregular cavities above the concrete pipe with an outer diameter of 0.6m and an inner diameter of 0.48m. Figure 9 shows a concrete pipe with an outer diameter of 1.2m and an inner diameter of 0.96m with some irregular cavities underneath. The spatial steps and time steps were 0.005 m and 0.01 ns, respectively, and the simulation iterations were 5000. Table 2 was the conductivity and relative permittivity of different materials in the model. The excitation source of the model was the same as above.

By analyzing Figure 10, the non-conformal model profile only shows one distinct reflected wave at the lower sides of the pipe. The profile of the conformal model shows that there are two hyperbolic reflection waves on the upper and lower sides of the pipe, respectively, and the reflected waves of the irregular cavity disease above the pipe are incredibly apparent. As shown in Figure 11, both the non-conformal and conformal model profiles clearly show the reflected waves at the top and bottom of the pipe, but the non-conformal model profile only shows the reflected waves from the cavity damage on the left side, while the conformal model profile completely shows the reflected waves distributed on both sides of the cavity damage. As can be seen in Figures 10 and 11, the circular pipe grid points are more accurately processed using the conformal grid technique.”

6. The Eqs. must be numbered in lines 100, 107, 119, 160, 166, 183.

Author's Response:

We appreciate the reviewer’s comment,and we have numbered in the above equations. Please refer to the cover letter for details.

7. Please check the paper for English editing and typos.

Author's Response:

We appreciate the reviewer’s comment,and we have carefully checked the paper for English editing and typos.

Reviewer 4 Report

  1. Figure 6 needs to be improved. all lines (FDTD, ADI-FDTD) look identical. Please do something.
  2. In chapter 4 it will be good to explain the cause of the discrepancy between simulated and measured results in figure 20. 

Author Response

1. Figure 6 needs to be improved. all lines (FDTD, ADI-FDTD) look identical. Please do something.

Author's Response:

We appreciate the reviewer’s comment, we have made corresponding modifications to Figure 6, as follows:

(You can find this image in your cover letter)

Figure 6. Comparison of Ez field distribution between ADI-FDTD and FDTD simulation.

2. In chapter 4 it will be good to explain the cause of the discrepancy between simulated and measured results in figure 20.

Author's Response:

We agree with the reviewer’s comment,we explain the reason for the discrepancy between the simulated and measured results in Figure 20 in line 360 on page 18. As follow:

It can be seen from Figure 20 that the simulated waveform obtained by the algorithm in this paper is in good agreement with the measured waveform in terms of amplitude and delay, with differences only in details. This is mainly due to the fact that we consider the soil layer as a homogeneous and isotropic medium, which leads to the difference between the assumed dielectric constant of the material and the real dielectric constant of the material during the simulation. This example verifies the applicability and effectiveness of the proposed algorithm for practical engineering inspection.”

Reviewer 5 Report

The manuscript presents the method of the GPR forward model integrated with the conformal ADI-FDTD approach. Mathematical models related to ADI-FDTD and procedures of numerical implementations are presented in addition to a brief introduction to conformal technology. The feasibility of the proposed method is demonstrated by several numerical applications and numerical results are compared to those by the field experiment. The topic is interesting and research outcomes could be potentially significant in practice. However, there is a lack of clear contribution of the proposed method to the research community and practice because ADI-FDTD, conformal technologies, and UPML boundary conditions in Section 2 had been developed and utilized. Most parts of numerical equations and derivations were also available in the literature. The manuscript should highlight the main contribution of the proposed method. Using the conformal technologies with the existing ADI-FDTD cannot be considered a novel method. The efficiency and accuracy for detecting underground pipeline and their defects that are the main objective of this research are not achieved by the new development. The review cannot recommend the current version of the manuscript be accepted for publication. There are questions and comments raised during the review.

  1. Introduction

In Line 65, it states that “…the CFL stability condition, and can employ a larger time step…” It is helpful for the reader to know more about the CFL stability issues and how the ADI-FDTD can address them.

In Line 74, it states that “the efficient and accurate GPR forward models are established…” Please elaborate on what development of the proposed method is new and different from the existing ADI-FDTD, UPML boundary conditions, and conformal technologies related to efficiency and accuracy.

  1. Methodology

in line 86, what does “TM” stand for? “Transverse magnetic”

The conformal technologies had been developed and proposed by other researchers such as:

  1. Chai, Tian Xiao, and Qing Huo Liu, "Conformal method to eliminate the ADI-FDTD staircase errors," in IEEE Transactions on Electromagnetic Compatibility, vol. 48, no. 2, pp. 273-281, May 2006, DOI: 10.1109/TEMC.2006.874084.

Please elaborate on the different aspects of the proposed method and its contribution to the research field.

  1. Numerical simulation

In Figure 7, there is a red arrow. However, it is hard to interpret its purpose of it. Please provide more explanation and how it can be read.

In Figure 7, it seems minor difference between Fig 7. (a) and (b) around the red arrow. What is a criterion considered that the result of Fig.7 (b) is better than the result shown in Fig. 7(a).  Can the problem domain be discretized further (e.g., mesh refinement/time and width) to get better resolutions? Does the mesh refinement affect the result as well? It would be helpful for readers to also find details regarding the numerical simulation and other parameters used as input variables.

The numerical example 1 shows various cases such as Non-conformal FDTD, conformal FDTD, and conformal ADI-FDTD. What about non-conformal ADI-FDTD? The comparison between non/conformal ADI-FDTD results and its discussion can be significant parts of the manuscript because it could show the advantage of the proposed method.

Author Response

1. In Line 65, it states that “…the CFL stability condition, and can employ a larger time step…” It is helpful for the reader to know more about the CFL stability issues and how the ADI-FDTD can address them.

Author's Response:

We thank the reviewers for their comments. In the ADI-FDTD method, a time step is divided into two sub-time steps. The solution of each sub-time step is conditionally stable after the split, but the solution after the combination of the two sub-time steps is unconditionally stable. CFLN is the ratio of the time steps of the ADI-FDTD algorithm to the FDTD algorithm that satisfies the stability condition. When CFLN is n, compared with the non-conformal FDTD algorithm, the time step of the conformal ADI-FDTD algorithm is increased to n times, and the number of iterations is reduced to 1/n. The time step can be increased by increasing the value of CFLN to improve computational efficiency. We can find these in lines 88 and 235 of the article, respectively.

2. In Line 74, it states that “the efficient and accurate GPR forward models are established…” Please elaborate on what development of the proposed method is new and different from the existing ADI-FDTD, UPML boundary conditions, and conformal technologies related to efficiency and accuracy.

Author's Response:

We thank the reviewers for their comments. Existing ADI-FDTD methods have high computational efficiency, but are still lacking in accuracy. Conformal technology makes the simulation more accurate, but cannot improve the computational efficiency. In this paper, the ADI-FDTD method and conformal technology are combined to improve the computational efficiency and computational accuracy. UPML boundary conditions provide good absorption. Based on the proposed algorithm, we obtained the GPR image features of the single-pipe model with cavity disease, the complex multi-pipe model and the complex underground structures, which provided a basis for further processing and interpretation of the GPR electromagnetic wave measured data.

3. in line 86, what does “TM” stand for? “Transverse magnetic”.

Author's Response:

We appreciate the reviewer’s comment. In the two-dimensional case, the electromagnetic field equations can be split into two independent sets of equations, one of which has only the Ez component. Such electromagnetic waves are called transverse magnetic waves and are denoted by TM. We modified line 83 of the article as follows:

“For transverse magnetic (TM) waves in two-dimensional lossy media, the Max-well equations [37] are expressed as”

4. The conformal technologies had been developed and proposed by other researchers. Please elaborate on the different aspects of the proposed method and its contribution to the research field.

Author's Response:

We appreciate the reviewer’s comment. At present, the conformal ADI-FDTD technique has been proposed by other researchers for simulating curved metal objects [1], microwave attenuation on coplanar waveguides (CPWs) with complex cross-sections [2], analyzing spiral inductors integrated on silicon substrates [3], describing optically controlled DR [4], and so on. Ground Penetrating Radar (GPR) is a fast and effective non-destructive testing method, which is widely used in the detection and positioning of pipelines and underground spaces. In this paper, the conformal ADI-FDTD technology is used as a numerical method to study the propagation law of GPR waves in complex underground structures, which improves the efficiency and accuracy of GPR in non-destructive testing, and provides a scientific basis for the application and promotion of GPR in practical engineering.

[1] Chai, Tian Xiao, and Qing Huo Liu, "Conformal method to eliminate the ADI-FDTD staircase errors," in IEEE Transactions on Electromagnetic Compatibility, vol. 48, no. 2, pp. 273-281, May 2006, DOI: 10.1109/TEMC.2006.874084.

[2] Chen H , Gan X . Computation of Microwave Attenuation on Coplanar Waveguide with Conformal ADI-FDTD Method[J]. International Journal of Infrared & Millimeter Waves, 2004, 25(2):291-299.

[3] Zheng H , Wu Y , Zhang K , et al. Wide-Band Modeling On-chip Spiral Inductors Using Frequency-Dependent Conformal ADI-FDTD Method[J]. IEEE Access, 2019, PP(99):1-1.

[4] Zheng H X . Modeling of optically controlled dielectric resonators using conformal ADI-FDTD method[C]// Microwave Conference Proceedings, 2005. APMC 2005. Asia-Pacific Conference Proceedings. IEEE, 2006.

5. In Figure 7, there is a red arrow. However, it is hard to interpret its purpose of it. Please provide more explanation and how it can be read.

Author's Response:

We appreciate the reviewer’s comment. We provide more explanation on how to understand the red arrows in Figure 7, as follows:

“Figure 7 shows the GPR profiles obtained by the non-conformal FDTD method, non-conformal ADI-FDTD method and the conformal ADI-FDTD method. We point out multiple reflections at the cavity with red arrows. As can be seen from Figure 7, the images of the non-conformal FDTD method and the non-conformal ADI-FDTD method are consistent. The conformal ADI-FDTD method has fewer multiple reflections than the non-conformal ADI-FDTD method. This indicates that the conformal ADI-FDTD method can effectively reduce the false reflection waves caused by conventional ladder approximation when simulating an underground circular cavity. Table 1 and Figure 7 show that the conformal ADI-FDTD algorithm can greatly improves the computational efficiency and also effectively improve the computational accuracy.”

6. In Figure 7, it seems minor difference between Fig 7. (a) and (b) around the red arrow. What is a criterion considered that the result of Fig.7 (b) is better than the result shown in Fig. 7(a). Can the problem domain be discretized further (e.g., mesh refinement/time and width) to get better resolutions? Does the mesh refinement affect the result as well? It would be helpful for readers to also find details regarding the numerical simulation and other parameters used as input variables.

Author's Response:

We are sorry to put the picture in the wrong position, the correct position after modification is shown in Figure 7. Figure 7(a) shows the GPR profile of the conformal ADI-FDTD method, Figure 7(b) shows the GPR profile of the non-conformal ADI-FDTD method, and Figure 7(c) shows the GPR profile of the non-conformal FDTD method. The red arrow locations represent multiple reflections. It can be seen that there are fewer multiple reflections in Figure 7(a) than in Figure 7(b) and Figure 7(c). Multiple reflections refer to the fact that a wave is reflected at one interface and then reflected more than once at another interface or the ground. Multiple reflections are a disturbance in the interpretation of reflected waves. Therefore, fewer multiple reflections help us interpret the true GPR data. We can obtain better accuracy in the problem domain by refining the width of the grid. But the calculation time of numerical simulation will increase and the calculation efficiency will decrease if the mesh size is too small. The size of the time step is related to the computational efficiency. The ADI-FDTD method in this paper improves the computational efficiency by increasing the time step.

(You can find pictures in the attachment.)

Figure 7. GPR B-scan images of the two-layer cavity model: (a) Obtained by conformal ADI-FDTD method; (b) Obtained by non-conformal ADI-FDTD method; (c) Obtained by non-conformal FDTD method.

7. The numerical example 1 shows various cases such as Non-conformal FDTD, conformal FDTD, and conformal ADI-FDTD. What about non-conformal ADI-FDTD? The comparison between non/conformal ADI-FDTD results and its discussion can be significant parts of the manuscript because it could show the advantage of the proposed method.

Author's Response:

We appreciate the reviewer’s comment. In order to compare and embody the advantages of the proposed algorithm. We have added a single-channel comparison image of the non-conformal ADI-FDTD method in Figure 6. The computational resource usage of the non-conformal ADI-FDTD method is added in Table 1. A profile image of the non-conformal ADI-FDTD method has been added to Figure 7. And revised the analysis about the computational efficiency and computational accuracy of the algorithm. As follow:

Table 1. Computational resource usage of different methods.

Method

Memory (Mb)

Δt (ns)

Iterations

Time (s)

Non-conformal FDTD

10.16

0.01

5000

1849

Conformal FDTD

10.20

0.01

5000

1897

Non-conformal ADI-FDTD

22.39

0.01

5000

4765

Conformal ADI-FDTD-CFLN=1

22.48

0.01

5000

4812

Conformal ADI-FDTD-CFLN=2

22.48

0.02

2500

2403

Conformal ADI-FDTD-CFLN=3

22.48

0.03

1667

1608

Conformal ADI-FDTD-CFLN=4

22.48

0.04

1250

1196

Conformal ADI-FDTD-CFLN=5

22.48

0.05

1000

961

“The time step of the conventional FDTD algorithm needs to satisfy the CFL stability condition. However, the ADI-FDTD algorithm is unconditionally stable and can save computation time by choosing a larger time step. Figure 6 shows a comparison of the Ez field distributions of the conformal ADI-FDTD with different CFLN, the non-conformal FDTD and the non-conformal ADI-FDTD. The CFLN is the ratio of the time step of the ADI-FDTD algorithm to that of the FDTD algorithm. It can be seen that the results of the non-conformal FDTD, the non-conformal ADI FDTD algorithm and the conformal ADI FDTD algorithm have good consistency in different time steps. Table 1 provides the calculation time of the different algorithms when simulating the circular cavity model, respectively. When CFLN is 5, compared with the non-conformal FDTD algorithm, the time step of the conformal ADI-FDTD algorithm is increased to 5 times, and the number of iterations is reduced to 1/5. At this point, the computation time of the conformal ADI-FDTD algorithm is 961 s and that of the non-conformal FDTD algorithm is 1849 s, saving about 48% of the time. Figure 7 shows the GPR profiles obtained by the non-conformal FDTD method, non-conformal ADI-FDTD method and the conformal ADI-FDTD method. We point out multiple reflections at the cavity with red arrows. As can be seen from Figure 7, the images of the non-conformal FDTD method and the non-conformal ADI-FDTD method are consistent. The conformal ADI-FDTD method has fewer multiple reflections than the non-conformal ADI-FDTD method. This indicates that the conformal ADI-FDTD method can effectively reduce the false reflection waves caused by conventional ladder approximation when simulating an underground circular cavity. Table 1 and Figure 7 show that the conformal ADI-FDTD algorithm can greatly improves the computational efficiency and also effectively improve the computational accuracy.”

(You can find pictures in the attachment.)

Figure 6. Comparison of Ez field distribution between ADI-FDTD and FDTD simulation.

(You can find pictures in the attachment.)

Figure 7. GPR B-scan images of the two-layer cavity model: (a) Obtained by conformal ADI-FDTD method; (b) Obtained by non-conformal ADI-FDTD method; (c) Obtained by non-conformal FDTD method.

Round 2

Reviewer 3 Report

All my comments were taken into account and necessary corrections were made. The article looks much better.

Author Response

(The authors gave the same response as above.)

Reviewer 5 Report

Thank you for the authors’ response and incorporation of the reviewer’s comments into the revised manuscript. The manuscirpt has been improved. However, the main concerns and questions in the previous review comments (please see below) were not answered and addressed in the revised manuscript.

“However, there is a lack of clear contribution of the proposed method to the research community and practice because ADI-FDTD, conformal technologies, and UPML boundary conditions in Section 2 had been developed and utilized. Most parts of numerical equations and derivations were also available in the literature. The manuscript should highlight the main contribution of the proposed method. Using the conformal technologies with the existing ADI-FDTD cannot be considered a novel method. The efficiency and accuracy for detecting underground pipeline and their defects that are the main objective of this research are not achieved by the new development.”

The conformal technique with ADI-FDTD/FDTD had been proposed by other researchers. The reviewer had a question about the novelty of the manuscript. The contribution of the manuscript is not clear because all equations and their derivations can be found in the literature. The manuscript was submitted as an “article” type, not an “industrial application.” Novelty and contributions of the proposed method to the research field should be stated and identified in the manuscript. Therefore, the reviewer still cannot recommend the current version of the manuscript be accepted for publication.

 Other comments are:

  1. The entire content including equations in Section 2.1 and 2.3 are identical to the following paper:

D.S. Feng, Q.W. Dai. GPR numerical simulation of full wave field based on UPML boundary condition of ADI-FDTD. NDT E 552 Int. 2011, 44, 495–504.

  1. Also using the conformal technique with ADI-FDTD is not new
  2. Chai, Tian Xiao, and Qing Huo Liu, "Conformal method to eliminate the ADI-FDTD staircase errors," in IEEE Transactions on Electromagnetic Compatibility, vol. 48, no. 2, pp. 273-281, May 2006, DOI: 10.1109/TEMC.2006.874084.

The manuscript does not highlight the novelty of the proposed method. It states that “The recursive formula of the electric field node is written as:” There are no details (steps, process, citation etc) provided and discussion on how that equation (Eq.(21)) is novel and new compared to literature.

  1. In the revised manuscript, the results from the non-conformal ADI-FDTD approach in Section 3.1 are almost identical to conformal ADI-FDTD. It states that “The conformal ADI-FDTD method has fewer multiple reflections than the non-conformal ADI-FDTD method”. How can we verify fewer multiple reflections are better results? Fewer multiple reflections could be correct ones. The validation and verification of results are missing
